# High-throughput screening identifies Aurora kinase B as a critical therapeutic target for Merkel cell carcinoma

Tara Gelb [1,6], Khalid A. Garman [1,6], Daniel Urban[2], Amy Coxon[1], Berkley Gryder [3,4], Natasha T. Hill[1], Lingling Miao[1], Tobie Lee[2], Olivia Lee [2], Sirisha Chakka[2], John Braisted[2], Jordan E. Jarvis [1], Rachael Glavin[1], Trisha S. Raj[1], Ying Xiao[1], Simone Difilippantonio[5], Amy Q. Wang[2], Min Shen[2], Ken Chih-Chien Cheng[2], Madhu Lal-Nag[2], Matthew D. Hall [2] & Isaac Brownell [1] ✉

Merkel cell carcinoma (MCC) is a rare, aggressive skin cancer. Most MCCs contain Merkel cell polyomavirus (virus-positive MCC; VP-MCC), and the remaining are virus-negative (VN-MCC). Immune checkpoint inhibitors are the first-line treatment for metastatic MCC, but durable responses are achieved in less than 50% of patients. To identify new treatments, we screen ~4,000 compounds for their ability to reduce MCC viability and demonstrate that VP-MCC and VN-MCC exhibit distinct response profiles. Aurora kinase inhibitors selectively reduce VP-MCC viability, with RNAi screening independently identifying AURKB as an essential gene for MCC survival, especially in VP-MCC. AZD2811, a selective AURKB inhibitor, induces mitotic dysregulation and apoptosis in MCC cells, with greater efficacy in VP-MCC. In mice, AZD2811 nanoparticles inhibit tumor growth and increase survival in both VP-MCC and VN-MCC xenograft models. Overall, our unbiased screens identify AURKB as a promising therapeutic target and AZD2811NP as a potential treatment for MCC.

Merkel cell carcinoma (MCC) is a rare, highly lethal neuroendocrine skin tumor characterized by frequent local recurrences and metastases[1–3]. Treatment with traditional chemotherapy is often ineffective with a median progression-free survival of ~3 months[4]. PD-(L)1 checkpoint inhibitors are effective for MCC; however, durable responses are seen in less than 50% of treated patients[5–13]. Thus, novel treatment strategies are needed.

Approximately 80% of MCC tumors are caused by Merkel cell polyomavirus (MCPyV)[2]. In virus-positive MCC (VP-MCC), MCPyV DNA expressing a truncated T antigen is integrated into the host genome but somatic gene mutations are uncommon. The remaining virus-negative MCC (VN-MCC) are associated with abundant UV-signature mutations affecting genes such as *TP53* and *RB1*[14,15]. To identify compounds that could be repurposed to treat MCC, we performed a high-throughput cell viability screen of ~4000 small molecules. This screen included ~2400 compounds in the NCATS Pharmaceutical Collection (NPC) that are approved by the FDA or other regulatory agencies around the world[16]. Additionally, we included a ~1900 compound library (Mechanism Interrogation PlatE: MIPE) comprised of oncology-focused and mechanistically annotated drugs[17,18]. These compounds

[1]Dermatology Branch, National Institute of Arthritis and Musculoskeletal and Skin Diseases, National Institutes of Health, Bethesda, MD 20892, USA. [2]National Center for Advancing Translational Sciences, National Institutes of Health, Rockville, MD 20850, USA. [3]Genetics Branch, National Cancer Institute, National Institutes of Health, Bethesda, MD 20892, USA. [4]Department of Genetics and Genome Sciences, Case Western Reserve University School of Medicine, Cleveland, OH 44106, USA. [5]Laboratory of Animal Sciences Program, Leidos Biomedical Research, Frederick National Laboratory for Cancer Research, Frederick, MD 21702, USA. [6]These authors contributed equally: Tara Gelb, Khalid A. Garman. ✉e-mail: isaac.brownell@nih.gov

were screened against three human VP-MCC cell lines and three human VN-MCC cell lines. To eliminate compounds that are generally toxic to cultured cells, we also screened four non-transformed control cell lines. Although VP-MCC and VN-MCC have distinct etiologies, limited attention has been given to comparing the pharmacogenomics of VP-MCC and VN-MCC.

Unsupervised hierarchical clustering demonstrated differential drug responses in MCC versus control cells, and between VP-MCC and VN-MCC. As VP-MCC are the most common subtype of MCC and the VP-MCC cell lines we used are more representative of native MCC tumors than the VN-MCC cell lines[19], we focused on drug target classes that specifically reduced the viability of VP-MCC, and inhibitors of aurora kinases (AURK) were among the top hits reducing VP-MCC viability.

Aurora kinase B (AURKB), along with Aurora kinases A (AURKA) and C (AURKC), make up the serine/threonine family of AURKs. AURKA and AURKB are responsible for cell-cycle regulation and have been implicated in tumorigenesis. Consequently, AURK inhibitors are being developed as anti-cancer agents[20–23]. Although AURKA and AURKB share sequence similarity they have distinct subcellular localizations and functions[24–26]. During mitosis, AURKB moves from centromeres, to the midzone, and ultimately the midbody where it regulates cytokinesis and has therefore been referred to as a "chromosome passenger"[25]. The chromosome passenger complex (CPC) consists of AURKB (the enzymatic component), survivin, the inner centromere protein (INCENP), and borealin[27]. The CPC is responsible for histone modification, control of the spindle-assembly checkpoint, regulation of cytokinesis[27,28], and ultimately accurate chromosome segregation[29]. As part of the CPC, survivin is phosphorylated by AURKB, helps localize the complex during mitosis[30], and is partially responsible for mitotic

progression[31]. Pharmacological inhibition of survivin has previously been reported to selectively reduce VP-MCC viability relative to VN-MCC[32], suggesting that AURKB targets may be critical for VP-MCC viability.

An arrayed druggable genome RNAi screen independently identified AURKB and its CPC binding partners, survivin and INCENP, as relevant MCC therapeutic targets. The selective AURKB inhibitor AZD2811 acted on-target in MCC cells using known assays of AURKB activity. Importantly, a nanoparticle formulation of AZD2811 (termed AZD2811NP) reduced MCC proliferation, decreased tumor volume, and extended survival in preclinical xenograft models of MCC. Overall, our study elucidated the distinct treatment landscape for VP-MCC and VN-MCC, led to the discovery of AURKB as a novel MCC therapeutic target, and identified a nanoparticle-based AURKB inhibitor as a new potential treatment for MCC.

## Results

### High throughput small molecule screen identification of MCC-selective compounds

We used high-throughput small molecule screening to identify agents that reduced MCC viability and could be candidates for drug repurposing to treat MCC. We screened 3908 compounds in the NPC and MIPE libraries. Each drug was screened at multiple doses against six human MCC cell lines (3 VP-MCC: WAGA, MKL-1, MKL-2, and 3 VN-MCC: MCC13, MCC26, UISO) and four non-transformed control cell lines (HACAT, HEK293T, CRL-7250, NIH-3T3) (Fig. 1A). We used area under the dose-response curve (AUC) analysis, which accounts for both potency and efficacy, to calculate the effectiveness of these compounds in reducing cell viability. Based on the distribution of small molecule activities and the expectation of sigmoidal dose-response

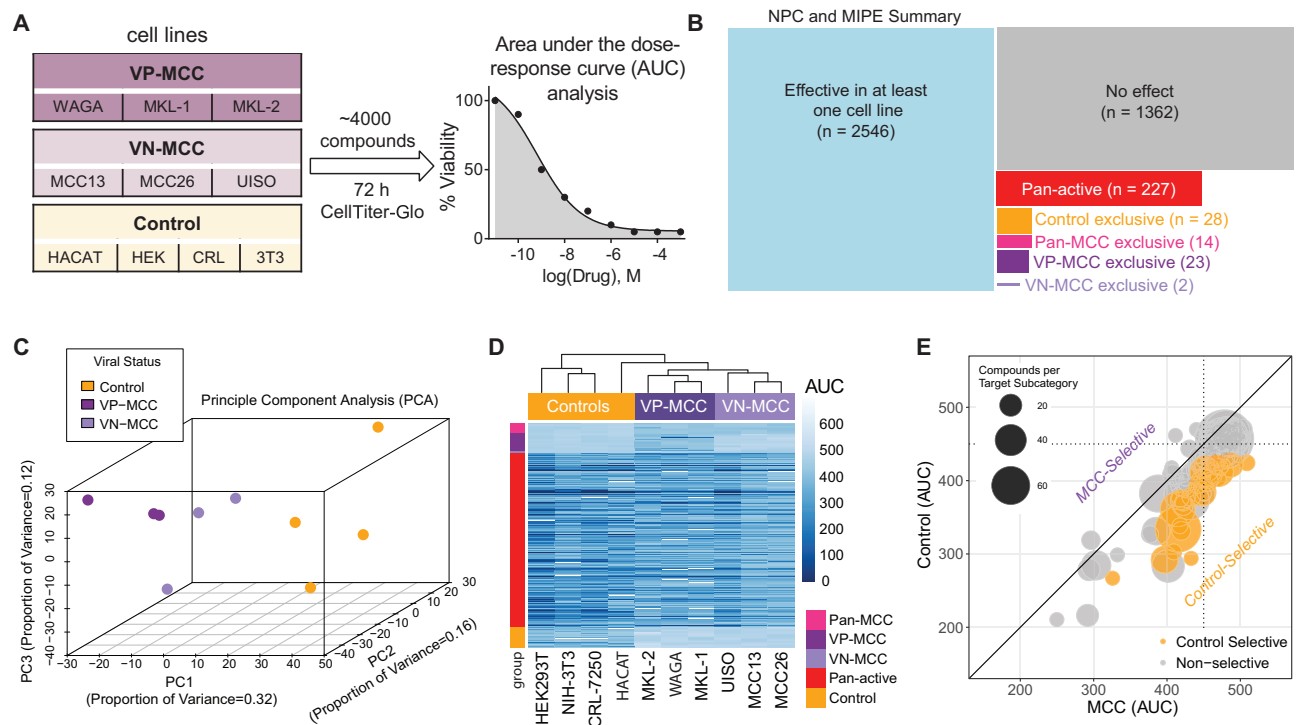

**Fig. 1 | VP-MCC and VN-MCC cell lines respond distinctly in a high-throughput small molecule screen. A** Three VP-MCC, three VN-MCC, and four immortalized control cell lines (n = 1) were screened against 3908 small molecules using viability as a readout. The area under the dose-response curve (AUC) was used to compare compounds. **B** Number of compounds that were effective (AUC < 450) in specific cell populations, size of the rectangle is proportional to the number of compounds. **C** Principal component analysis of the 2546 compounds that reduced viability in at least one cell line (AUC < 450). **D** Hierarchical clustering of

cell lines based on responses to 294 compounds with pan-active or cell-type exclusive responses. The heatmap shows AUC (darker blue indicates more potent and efficacious). **E** Bubble plot demonstrating mean AUC in MCC cell lines vs. mean AUC in control cell lines for compounds grouped by target subcategory. Orange bubbles indicate a selective reduction of control cell viability. Please refer to Supplementary Fig. S4 for VP-MCC and VN-MCC-selective target subcategories compared to control cell lines. n = number of experimental replicates for each cell line.

curves, an AUC of < 450 (mean of 0.25 quantile AUC for all 10 cell lines included in study = 449.6) was selected as a threshold for small molecule inhibitory activity (Supplementary Fig. S1).

Approximately one-third of the compounds tested (1362/3908) failed to reduce viability (AUC ≥ 450) in all the cell lines screened (Fig. 1B). Most compounds (2546/3908) effectively reduced viability (AUC < 450) in at least one of the cell lines screened (Fig. 1B). We performed a thorough evaluation of activity and selectivity of these compounds by employing two distinct analytical approaches. The first analysis classified compounds with an AUC < 450 into three primary groups: pan-active compounds reducing viability in all 10 cell lines (n = 227), control-selective compounds exclusively reducing viability of all 4 control cell lines without affecting MCC cell lines (n = 28), and MCC-selective compounds exclusively reducing viability in MCC cell lines without affecting control cell lines (n = 39) (Fig. 1B). The remaining 2252 compounds had activity that was not specific to one of the three primary cell groups. The 39 MCC-selective compounds were further subclassified into compounds reducing viability in all six MCC cell lines (n = 14), only in VP-MCC cell lines (and not VN-MCC or control cells) (n = 23), or only in VN-MCC cell lines (and not VP-MCC or control cells) (n = 2) (Fig. 1B and Supplementary Dataset 1). The second analytical method focused on identifying compounds with an "in vitro therapeutic window" by comparing average AUC values between control and MCC cell lines. Compounds were selected if the average AUC in control cells was higher than in MCC cells, with an average AUC difference > 50, p-value < 0.05, and AUC for MCC cell lines < 450 (Supplementary Fig. S2 and Supplementary Dataset 2). This analysis identified compounds that could possibly be repurposed to treat MCC as they were more effective in MCC cells despite some activity in control cells, including those selectively effective in all MCC cell lines (n = 18), VP-MCC cells (n = 98), and VN-MCC cells (n = 15).

Principal component analysis (PCA) (Fig. 1C) and hierarchical clustering (Fig. 1D and Supplementary Fig. S3) demonstrate that MCC and control cells respond distinctly to the screened compounds. Moreover, distinct VP-MCC and VN-MCC clusters indicate that their divergent pathophysiologies[14,15] correlate with differential drug response profiles.

As individual compounds demonstrating efficacy in MCC might do so due to off-target effects, we sought to enrich for drugs with on-target, mechanistically relevant effects by identifying compound targets where multiple compounds showed activity. To help prioritize drug targets, compounds with functional annotations (n = 1910)[33] were grouped by target subcategories based on the mechanism of action of the compounds (156 target subcategories, Supplementary Dataset 4). For each target subcategory, the average AUC of MCC cell lines (*MCC(AUC)*) was compared to the average AUC of control cell lines (*Control(AUC)*). No target subcategories were significantly associated with selective activity against MCC cells relative to control cells (Fig. 1E), indicating the pan-MCC selective compounds in the screen did not share functional targets. The lack of pan-MCC therapeutic targets was not unexpected considering the divergent pathophysiology and the distinct drug responses of VP-MCC and VN-MCC cell lines. Therefore, we analyzed the two MCC subtypes (VP-MCC and VN-MCC) independently.

**Identification of VP-MCC and VN-MCC-selective drug targets and compounds.** Comparison of VP-MCC or VN-MCC responses to control cells identified agents that were selective for each MCC subtype (Supplementary Fig. S2). Grouping compounds by target subcategory[33], we identified four targets enriched among the VP-MCC selective agents, but no targets were selective for VN-MCC cells (Supplementary Fig. S4). To further investigate the differences between VP-MCC and VN-MCC we compared them directly to each other.

To identify targets critical for either VP-MCC or VN-MCC viability, we compared the average *VP-MCC(AUC)* to the average *VN-MCC(AUC)*

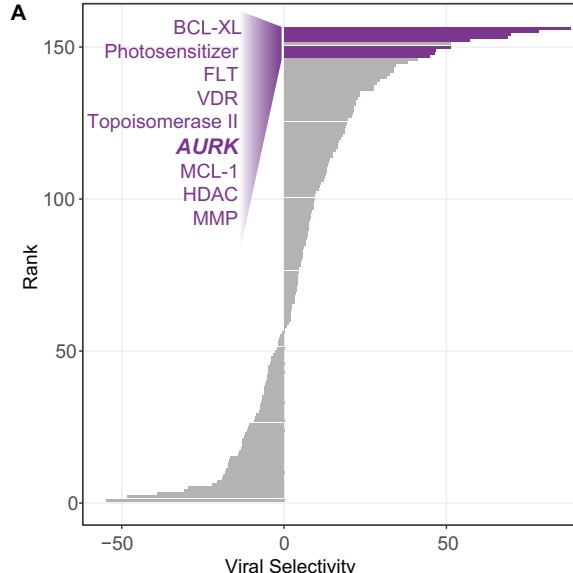

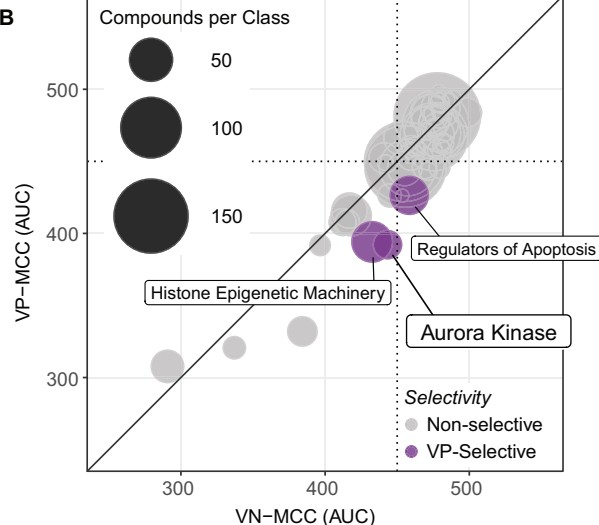

**Fig. 2 | Identification of VP-MCC selective targets. A** Drug target subcategories ranked by difference in mean AUC in VP-MCC and VN-MCC cell lines. Dark-purple bars indicate VP-MCC selectivity (AUC difference > 40, p < 0.1, AUC < 450 in all VP-MCC cell lines screened). P-values: BCL-XL: (p = 0.0007; n = 6), photosensitizer: (p = 0.037; n = 4), FLT: (p = 0.019; n = 4), VDR: (p = 0.094; n = 3), Topoisomerase II: (p = 0.069; n = 21), AURK: (p = 0.059; n = 18), MCL-1: (p = 0.043; n = 4), HDAC: (p = 0.034; n = 25), MMP: (p = 0.0024; n = 11). **B** Bubble plot demonstrating mean VN-MCC AUC vs. mean VP-MCC AUC for each compound class (size of bubble indicates number of agents in compound class). Purple bubbles highlight compound classes that selectively reduced VP-MCC viability (AUC difference > 30, p < 0.1, AUC < 450 in all VP-MCC cell lines screened). P-values: aurora kinase inhibitors: (p = 0.059; n = 21), regulators of histone epigenetic machinery: (p = 0.043; n = 44), regulators of apoptosis: (p = 0.013; n = 39). A two-sided t-test was used to calculate the p-values. n = number of compounds.

for each target subcategory. There were nine targets, including aurora kinases (AURKs), that when inhibited selectively reduced VP-MCC viability relative to VN-MCC viability (AUC difference > 40, p < 0.1, and AUC < 450 for all VP-MCC cell lines, Fig. 2A). A subset of these VP-MCC selective target subcategories were also associated with reduced VP-MCC viability relative to controls (Supplementary Fig. S4A). There were no groups of drugs with the same target that were selective for VN-MCC relative to VP-MCC (Fig. 2A). We focused on validating and

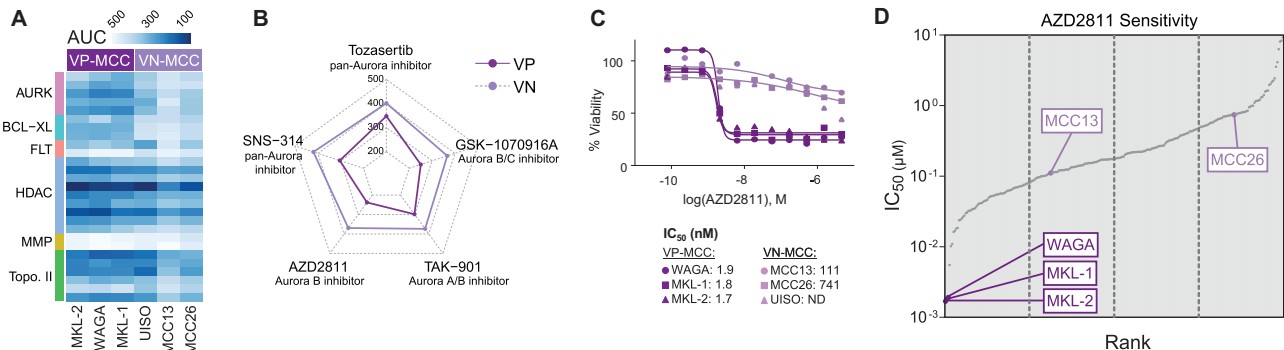

**Fig. 3 | Follow-up small molecule screen validated aurora kinase inhibitors as effective in VP-MCC. A** Heatmap of select drug responses in follow-up screen that validated high-priority target subcategories. Please refer to Supplementary Fig. S5A for a heatmap of all 128 compounds included in the follow-up screen (n = 3). **B** Radar plot demonstrates a smaller mean AUC for AURK inhibitors in VP-MCC than VN-MCC cell lines. **C** Representative dose-response curves and IC$_{50}$ values of AZD2811 in MCC cell lines. Please refer to Supplementary Fig. S6A, B for dose-response curves and IC$_{50}$ values of additional AURK inhibitors. **D** The IC$_{50}$ value of AZD2811 (μM) ranked across 246 cancer cell lines included in the Dependency Map (gray) and MCC cell lines from our high-throughput screen (purple). AZD2811 was more potent in MKL-2, MKL-1, and WAGA (VP-MCC cell lines) than > 99% of other cancer cell lines included in this database. n = number of experimental replicates.

developing drugs that selectively reduced VP-MCC viability because VP-MCC is more common than VN-MCC, the VP-MCC cell lines we used are more representative of native MCC tumors than the VN-MCC cell lines[19], and more VP-MCC candidate compounds were identified in the screen. In this approach we used the VN-MCC cell lines as closely-related controls/comparators to better identify targets specific to VP-MCC.

To further integrate the functional mechanisms of effective inhibitors, we combined the 156 target subcategories into broader compound classes that include compounds based on their general pharmacological or therapeutic function (56 compound classes, Supplementary Dataset 4). Comparison of average VP-MCC and VN-MCC AUCs revealed that three compound classes (AURK inhibitors, regulators of intrinsic apoptosis, and histone epigenetic machinery regulators) selectively reduced VP-MCC viability relative to VN-MCC (Fig. 2B). These results suggest that VP-MCC viability is dependent on proteins targeted by these compound classes.

**Follow-up small molecule screens demonstrate replicability of primary screen.** To validate hits from our initial screen, we re-tested a subset of compounds (n = 128) in a follow-up screen performed in triplicate (Fig. 3A and Supplementary Fig. S5A). These compounds were selected based on their performance in the primary screen or notable reports of their effect on MCC from the literature. Independent replicate runs (n = 3) for each MCC cell line clustered together in a PCA, demonstrating the precision of our screening method (Supplementary Fig. S5B). Plotting the primary screen results for the 128 compounds in the same PCA showed a consistent offset for each cell line, suggesting a systematic batch effect between the primary and follow-up screens. Nonetheless, there was a strong correlation between the AUC values in the primary and follow-up screens for all six MCC cell lines (Supplementary Fig. S5C), validating the screening results. In the follow-up screen, VP-MCC and VN-MCC cell lines once again formed separate clusters (Supplementary Fig. S5A, B), confirming divergent drug responses in the two MCC subtypes. The compounds validated in the follow-up screen were members of select target subcategories that were prioritized based on effect size and clinical potential of available drugs. Among the validated target subcategories, we selected AURK for further study and preclinical development.

**Follow-up small molecule screens support AURKs as a high priority target for VP-MCC.** Having identified AURKs as VP-MCC selective targets (Fig. 2A), and AURK inhibitors as a candidate compound class

(Fig. 2B) in our primary high-throughput screen, we re-tested five AURK inhibitors (AZD2811, GSK-1070916A, SNS-314, TAK-901, and tozasertib) in triplicate, and found that all five consistently and potently reduced VP-MCC viability (Fig. 3B, C and Supplementary Fig. S6A, B). IC$_{50}$ values in the VP-MCC cell lines ranged from 1.7–40.4 nM for all five AURK inhibitors tested (Fig. 3C and Supplementary Fig. S6B). In contrast, AURK inhibitors were notably less potent against VN-MCC cells. For responses that could be fit to a sigmoidal dose-response curve, IC$_{50}$ values in the VN-MCC cell lines ranged from 111 nM–24 mM (Fig. 3C and Supplementary Fig. S6B). However, extending the treatment time from 72 h to 144 h notably increased both the in vitro potency and efficacy of AZD2811 in VN-MCC cell lines (Supplementary Fig. S6A, B). Using publicly available IC$_{50}$ data in the Dependency Map (DepMap)[34] and Genomics of Drug Sensitivity in Cancer (GDSC)[35] databases, we observed the potencies of the AURK inhibitors AZD2811 (DepMap, Fig. 3D) and tozasertib and GSK-1070916A (GSDC, Supplementary Fig. S6C) were higher in VP-MCC cell lines than in hundreds of other human cancer cell lines. Despite the limitations of the potency estimates reported in these databases, when combined with our results they support AURK inhibitors having relatively high potency in VP-MCC.

**RNAi screening independently identifies aurora kinase B (AURKB) as a MCC therapeutic target.** To independently identify monogenic dependencies of MCC cell viability, we performed an arrayed druggable genome RNAi screen in VP-MCC (MKL-2) and VN-MCC (MCC26) cells. These cell lines were selected because they demonstrated higher transfection efficiency than other MCC lines. The effect of knockdown of 10,418 individual genes on cell viability was determined and the data were normalized to a non-targeting siRNA negative control. AURKB knockdown robustly reduced viability of MKL-2 cells (VP-MCC) and to a lesser degree decreased the viability of MCC26 cells (VN-MCC) (Supplementary Dataset 3 and Fig. 4A, B). In contrast, AURKA knockdown only reduced MKL-2 viability and AURKC knockdown did not alter MKL-2 or MCC26 viability (Fig. 4A). In follow-up knockdown studies, AURKB siRNA efficiently reduced protein levels (Fig. 4C, D) and viability (Fig. 4E) in VP-MCC (WAGA and MKL-1) and VN-MCC (MCC13 and UISO) cell lines. Notably, the baseline protein levels of AURKB were similar in both VP-MCC and VN-MCC (Fig. 4C). These results demonstrate that although AURK inhibitors are more potent in VP-MCC, AURKB is required for the viability of both VP-MCC and VN-MCC. Moreover, through the RNAi screen we discovered that AURKB, which is expressed in human MCC tumors (Supplementary Fig. S7), is likely the relevant AURK isoform required to sustain MCC viability.

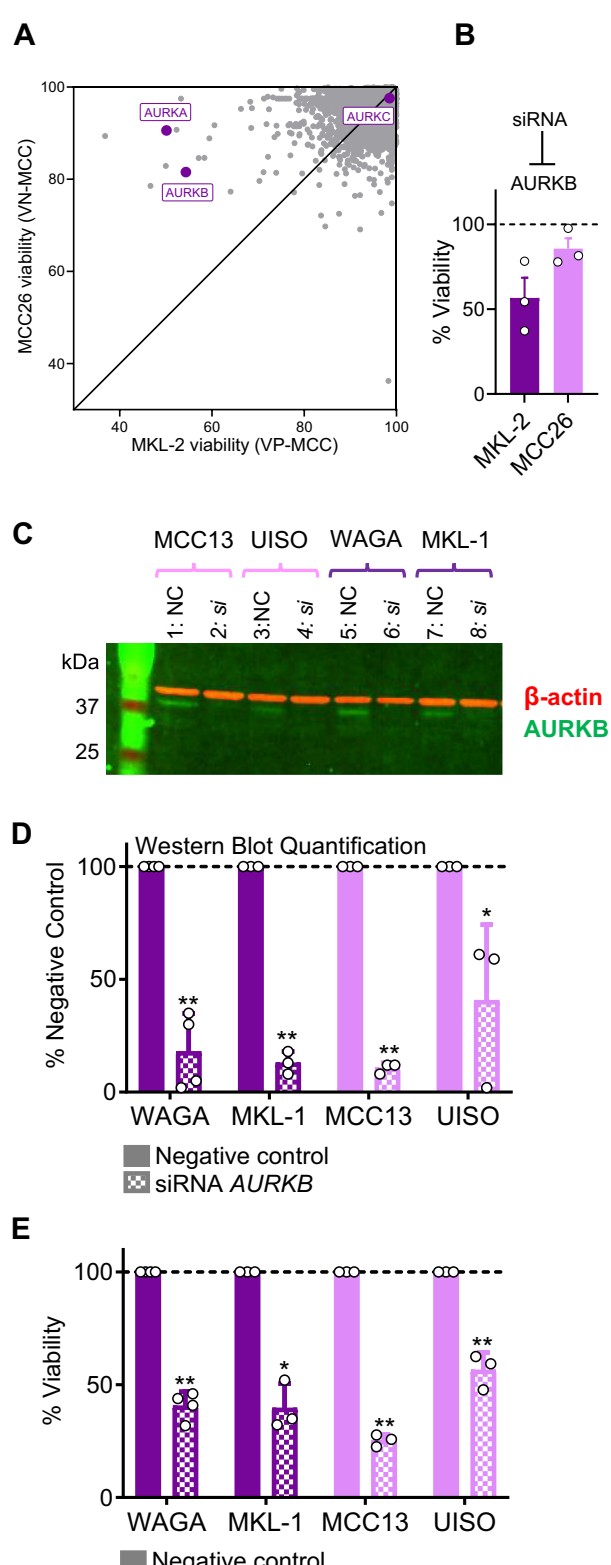

**Fig. 4 | AURKB knockdown reduces MCC viability. A** RNAi druggable genome screen demonstrating mean viability reduction following knockdown in MKL-2 (VP-MCC) vs. MCC26 (VN-MCC) cells (n = 3). All viabilities > 100 set to 100. Please refer to Supplementary Fig. S8: knockdown of both INCENP (n = 3) and survivin (n = 5) reduce VP-MCC, but not VN-MCC viability. **B** AURKB knockdown reduces MKL-2 and MCC26 viability relative to negative control (NC) (n = 3). Mean viability from 3 different siRNAs, error bars represent SEM. **C** Representative Western Blot demonstrating AURKB knockdown with siRNA (si), but not NC in MCC cell lines (n = 3). Red: β-actin loading control, green: AURKB. **D** Western Blot quantification demonstrating efficient AURKB siRNA knockdown relative to NC in WAGA (p = 0.001; n = 4), MKL-1 (p = 0.00003; n = 3), MCC13 (p = 0.00009; n = 3), and UISO (p = 0.045; n = 3) cell lines as measured by one-tailed two-sample t-test with unequal variance. AURKB intensity was normalized to β-actin intensity from identical lane, and means from at least 3 independent experiments are shown, with error bars representing SD. **E** AURKB knockdown significantly reduced viability of WAGA (p = 0.0003; n = 4), MKL-1 (p = 0.0105; n = 3), MCC13 (p = 0.0004; n = 3), and UISO (p = 0.0094; n = 3) cells relative to NC as measured by one-sample t-test. Data points are means from at least 3 independent experiments, with error bars representing SD. A and B, n = number of siRNAs per gene; C, D and E, n = number of experimental replicates.

AURKB is involved in the phosphorylation and regulation of the microtubule destabilizing protein stathmin-1[36]. Stathmin-1 is overexpressed in various cancers and is correlated with a poor prognosis[37,38]. Others have reported that the protein level of stathmin-1 is regulated by the small-T antigen of the Merkel cell polyomavirus (MCPyV-sT)[39]. Expression of MCPyV-sT in HEK293 or MCC13 cells led to robust upregulation of stathmin-1 and subsequent increases in motility and migration[39]. Consistent with this, we found that stathmin-1 protein expression is consistently higher in VP-MCC than VN-MCC cell lines (Supplementary Fig. S8D, E), suggesting that elevated stathmin-1 mediated microtubule destabilization may account for some of the disproportional impacts of AURKB inhibitors in VP-MCC.

**The AURKB inhibitor AZD2811 reduces Histone 3-Ser 10 phosphorylation (H3Ser10) and induces mitotic dysregulation in MCC.** Having identified the B isoform as the relevant AURK target in MCC, we employed biochemical profiling to determine the AURK inhibitor with the highest potency against AURKB relative to other isoforms. AZD2811 was the only compound that was most potent against AURKB, whereas four other AURK inhibitors most potently inhibited AURKA (Supplementary Fig. S9). Therefore, we used the AURK inhibitor AZD2811 in further studies. AZD2811 was also selected for further development owing to its availability as a nanoparticle formulation with favorable pharmacokinetics in early-phase human clinical trials[40–42].

Given that AURKB is known to phosphorylate Histone 3 on Ser 10 (H3Ser10), a crucial step for chromosome condensation and mitosis[25,43], we aimed to validate this on-target effect of AZD2811[44] in MCC. Treating both VP-MCC (WAGA, MKL-2, MKL-1) and VN-MCC (MCC13 and MCC26) cell lines with AZD2811, we observed a dose-dependent reduction in H3Ser10 phosphorylation at 24 h in WAGA, MKL-2, MKL-1, and MCC13, and at 72 h in MCC26 (Fig. 5A, B).

With the on-target effect of AZD2811 on H3Ser10 confirmed and the known role of AURKB in regulating cytokinesis and chromosome segregation[45], we hypothesized that treatment of MCC cells with AZD2811 would induce late phase cell cycle dysregulation. Cell cycle analysis by flow cytometry revealed that, compared to the VP-MCC cell lines (WAGA, MKL-2, MKL-1), the VN-MCC cell lines MCC13 and MCC26 had a higher frequency of G2 cells, polyploidy, and aneuploidy at baseline (Fig. 5C, D). Treatment with AZD2811 at a concentration of 30 nM induced G2 arrest in WAGA, MKL-2, MKL-1, and MCC13 at 24 h as evidenced by the increased proportion of G2 cells (4 N) (Fig. 5C, D). In MCC26 cells there was an increase in polyploid cells (> 4 N) observed at 24 and 72 h, suggesting mitotic exit without cytokinesis. Thus, in all

**AURKB signaling partners in the CPC also maintain VP-MCC viability.** AURKB is part of the chromosomal passenger complex (CPC) with three other proteins: survivin, INCENP, and borealin. Consistent with prior reports of MCC dependency on survivin[32], we found that pharmacological inhibition of survivin or its siRNA knockdown more potently reduced VP-MCC viability relative to VN-MCC (Supplementary Fig. S8A, C). We found similar results after siRNA knockdown of INCENP (Supplementary Fig. S8F), further suggesting that AURKB and the CPC are required for VP-MCC viability.

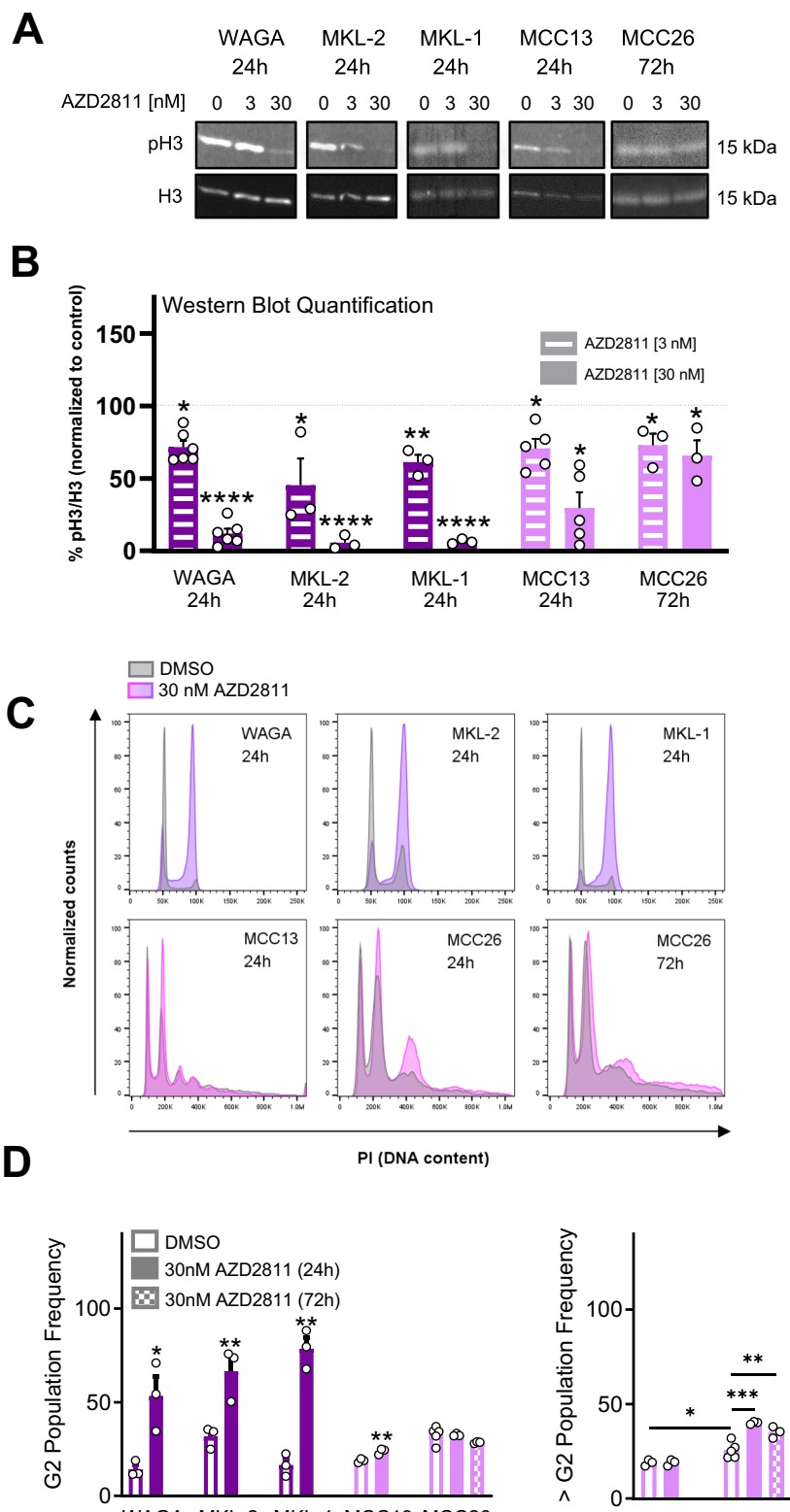

MCC cell lines AZD2811 blocked cytokinesis resulting in either G2 arrest or increased polyploidy.

As cells with ≥ 4 N are susceptible to mitotic catastrophe[46], we speculated that this may drive cell death in MCC cells treated with AZD2811. Mitotic catastrophe occurs in cells unable to complete mitosis, characterized by the appearance of micronucleation, large multinucleated cells, and eventual cell death[47]. In WAGA, MKL-1, and MCC26 cells, 72-h treatment with AZD2811 resulted in cell enlargement, micronucleation, multinucleation with centrosome amplification, and apoptosis (Supplementary Figs. S10 and S11). At baseline, a portion of MCC13 cells demonstrated enlarged cell size, micronucleation, and multinucleation with centrosome amplification, suggesting these cells exist in a state of mitotic dysregulation. AZD2811 treatment for 72 h induced subjective nuclear deformities with fusion of the multiple nuclei but had a negligible effect on the average cell size, percent of cells with centrosome amplification, and the induction

**Fig. 5 | Aurora kinase B inhibitor, AZD2811 reduces histone 3 Serine 10 (H3Ser10) phosphorylation and induces mitotic dysregulation in MCC cells.** **A** Representative Fluorescent Western Blot of total histone 3 expression and pH3-Ser10. MCC cells were treated with DMSO, 3 nM, or 30 nM AZD2811 for 24 h (72 h for MCC26). Images were cropped to improve clarity (n = 3). **B** Quantification of pH3-Ser10 protein expression normalized to total histone 3 relative to DMSO control treatment from three independent experiments, with error bars representing SEM. 30 nM of AZD2811 significantly reduced pH3-Ser10 in WAGA (3 nM: p = 0.002, 30 nM: p < 0.0001; n = 6), MKL-2 (3 nM: p = 0.097, 30 nM: p = 0.00072; n = 3), MKL-1 (3 nM: p = 0.002, 30 nM: p < 0.0001; n = 3), MCC13 (3 nM: p = 0.043, 30 nM: p = 0.011; n = 5), and MCC26 (3 nM: p = 0.026, 30 nM: p = 0.031; n = 3) cells, as measured by two-tailed, two-sample, unpaired t-test. **C** Representative flow

cytometry propidium iodide (PI) intensity histogram traces following treatment with 30 nM AZD2811 demonstrating a G2/M cell cycle arrest at 24 h in WAGA, MKL-2, MKL-1, and MCC13 and increased polyploidy ( > G2) at 24 and 72 h in MCC26 (n = 3). **D** AZD2811 24-h treatment (30 nM) significantly increased the proportion of cells in G2 in WAGA (p = 0.022; n = 3), MKL-1 (p = 0.0008; n = 3), MKL-2 (p = 0.016; n = 3), MCC13 (p = 0.0056; n = 3), but not MCC26 cells (p = 0.149; n = 3) as measured by two-sample, one-tailed unpaired t-test. AZD2811 treatment (30 nM) significantly increased the proportion of polyploid cells ( > G2) in MCC26 at 24 h (p = 0.0006; n = 3) and 72 h (p = 0.00759; n = 3). Values are means from at least 3 independent experiments with error bars representing SEM. n = number of experimental replicates.

of apoptosis (Supplementary Figs. S10 and S11). However, extending AZD2811 treatment to 144 h produced a significant increase in cell size and increasing the concentration from 30 nM to 300 nM significantly increased the percent of MCC13 cells with centrosome amplification (Supplementary Fig. S10D, E). Increasing the concentration and treatment time of MCC13 cells did not meaningfully induce apoptosis (Supplementary Fig. S11A). Taken together, AZD2811 produced morphological and chromosomal changes consistent with the induction of mitotic catastrophe that resulted in high levels of apoptosis in the VP-MCC (WAGA and MKL-1) cells, correlating with AZD2811's high potency in these cell lines. In contrast, MCC13 and MCC26 VN-MCC cell lines demonstrated baseline evidence of dysregulated mitosis (increased G2, polyploidy, aneuploidy, micronucleation, increased cell size, and/ or multinucleation with centrosome amplification) that was variably impacted by AZD2811 treatment. AZD2811-induced apoptosis is weaker in MCC26 and lacking in MCC13, which correlates to the lower potency of AZD2811 observed in VN-MCC cell lines.

As VP-MCC cell lines require sustained expression of MCPyV T antigens[2], the increased potency of AZD2811 in VP-MCC over VN-MCC could also be due to targeting T antigen expression. However, treatment with AZD2811 did not change the expression of MCPyV-LT in VP-MCC cell lines (Supplementary Fig. S12).

**Inhibition of aurora kinase B reduces tumor burden and extends survival in preclinical mouse models of MCC.** To determine if AURKB inhibitors could be repurposed to treat MCC in a more physiologically relevant model, we tested AZD2811NP, a nanoparticle formulation of AZD2811, in xenograft mouse models of MCC[19]. MKL-1 (VP-MCC) or MCC13 (VN-MCC) cells were engrafted into nude mice (n = 10 animals/ group) and bi-weekly treatment was initiated once the average tumor volume reached 100 mm³. Animals received either 25 mg/kg AZD2811NP or placebo nanoparticles for four weeks (Fig. 6A) and were monitored for a total of either 242 (MKL-1) or 114 days (MCC13).

During treatment, AZD2811NP significantly slowed the growth of MKL-1 tumors relative to placebo (p < 0.0001 as measured by linear regression of growth curves, Supplementary Fig. S13A) without causing a reduction in body weight (Supplementary Fig. S13C) or other evidence of toxicity. Treated tumors shrank from their baseline size (slope of linear regression of growth curve = -2.4, 95% confidence interval = -3.1 to -1.7) (Fig. 6B, F and Supplementary Fig. S13A). After 4 weeks of treatment, the average MKL-1 placebo treated tumor was 616 mm³ whereas the average AZD2811NP treated MKL-1 tumor was only 32.5 mm³ (p < 0.005 as measured by one-tailed, two-sample t-test). Even after discontinuation of treatments, AZD2811NP treated MKL-1 xenografted mice maintained benefit, with a median survival of 229 days relative to 63.5 days in the placebo group (3.6-fold increase, Fig. 6D). Moreover, none of the MKL-1 AZD2811NP treated mice died of tumor burden in 150 days of monitoring (Fig. 6D). In contrast, 90% of placebo treated mice were dead within 150 days (Fig. 6D). Tissue collected from the treated MKL-1 xenografts contained no viable tumor, preventing meaningful analysis. In an attempt to study the mechanism of AZD2811NP in vivo, we generated more MKL-1 xenograft bearing

mice and treated with AZD2811NP for only 10 days. Xenograft tissue (n = 3) collected at 10 days of treatment also contained no viable tumor, emphasizing how in vivo responses to AZD2811NP were both rapid and robust in existing MKL-1 tumors. Overall, these studies identify AZD2811NP as a possible novel treatment option for VP-MCC.

AZD2811 and other AURK inhibitors were less potent in reducing VN-MCC viability in vitro (Fig. 3 and Supplementary Fig. S6) suggesting that AZD2811NP may be ineffective against MCC13 (VN-MCC) xenografts. However, AZD2811NP significantly slowed MCC13 (VN-MCC) tumor growth relative to placebo (Fig. 6C, G and Supplementary Fig. S13B) and extended median survival from 31.5 days in placebo-treated mice to 84 days (2.7-fold increase, Fig. 6E) without causing weight loss (Supplementary Fig. S13D) or other adverse toxicities. To understand the mechanism of action of AZD2811NP in MCC13 xenograft tumors, we generated new tumor-bearing mice that we treated for 10 days. Tumors collected at 10 days showed reduced H3Ser10 phosphorylation and reduced proliferation as indicated by Ki-67 expression. Consistent with in vitro results, AZD2811NP treatment did not induce a significant increase in the apoptosis marker cleaved caspase-3 (Supplementary Fig. S14). Thus, AURKB inhibition appears to be cytostatic to MCC13 xenograft tumors.

To determine if the 25 mg/kg dose of AZD2811NP produced a tissue concentration high enough to reduce VN-MCC viability, we conducted a pharmacokinetic study in tumor-free mice. Mice were treated with a single intravenous injection (IV) of 25 mg/kg AZD2811NP. Plasma and liver samples were collected at 10 time-points (ranging from 0.168-504 h). The maximum concentrations of AZD2811 ($C_{max}$) in the plasma and liver were 276 µg/mL (~0.54 mM) and 97.1 µg/g (~0.19 mM), respectively (Supplementary Fig. S13E–G). The maximum tissue concentrations were observed at 7 and 24 h and the half-life ($t_{1/2}$) was 149 and 61 h, respectively (Supplementary Fig. S13E–G), suggesting that twice-weekly dosing would sustain blood and tissue concentrations higher than the maximum concentration of AZD2811 we tested in vitro (Fig. 3C) or used in our functional studies (Fig. 5 and Supplementary Figs. S10 and S11). Taken together, these xenograft studies strongly support the identification of AURKB as a relevant therapeutic target and justify testing AZD2811NP as a possible therapy for patients with advanced VP-MCC or VN-MCC.

## Discussion

The first line of treatments for metastatic MCC are anti-PD-(L)1 immune checkpoint inhibitors (ICIs). ICIs show impressive objective response rates but unfortunately produce durable responses in less than 50% of patients[7–12]. Some patients cannot receive ICIs due to autoimmune disease or immunosuppression. Therefore, we used unbiased, high-throughput screening techniques to identify new targeted treatments, which could be used alone or in combination with checkpoint inhibitors, for this deadly cancer. Intersectional and complementary small molecule and RNAi screening allowed us to identify AURKB as a therapeutic target and develop the nanoparticle AURKB inhibitor AZD2811NP as a novel treatment approach for MCC.

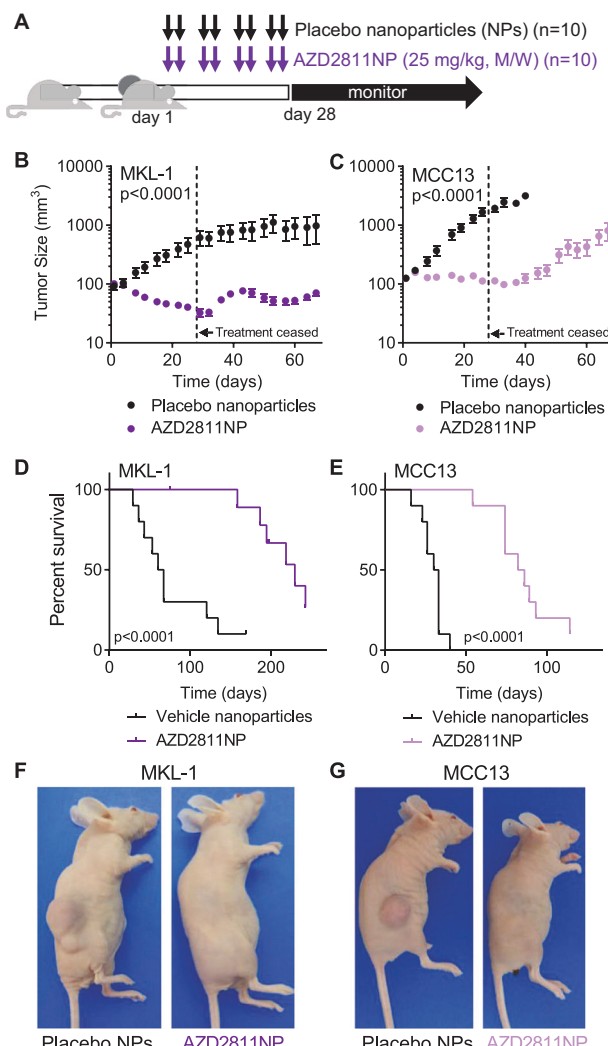

**Fig. 6 | Inhibition of Aurora kinase B shrinks MCC tumors in preclinical mouse models of MCC. A** Schematic of xenograft tumor treatment protocol. Mice were subcutaneously injected with either MKL-1 or MCC13 cells (n = 10). Once the average tumor size reached 100 mm³ animals were treated with intravenous AZD2811NP (25 mg/kg, twice weekly) or placebo nanoparticles for 4 weeks. **B, C** Rate of tumor growth, as calculated by linear regression, was significantly reduced by AZD2811NP treatment relative to placebo. P-values on the slope comparing vehicle and AZD2811NP were MKL-1 (p < 0.0001) and MCC13 (p < 0.0001). Data points represent average tumor volumes, with error bars representing SEM (n = 10). Please refer to Supplementary Fig. S13: AZD2811NP significantly slowed MKL-1 and MCC13 tumor growth relative to placebo during treatment period as measured by linear regression. Treatment with AZD2811NP did not reduce body weight in mice xenografted with MKL-1 (p = 0.71) or MCC13 (p = 0.9). **D, E** Kaplan Meier survival estimates showing that AZD2811NP extends survival of mice with MKL-1 (p < 0.0001) or MCC13 (p < 0.0001) xenograft tumors as calculated by Log-rank (Mantel-Cox) test in GraphPad Prism (n = 10). **F, G** Representative photographs of mice (MKL-1: day 28; MCC13: day 22) labeled with treatment. **A, D, E, F,** and **G,** n = number of animals; **B** and **C,** n = number of tumors.

MCC is difficult to treat, possibly because there are two distinct subtypes. VP-MCC is driven by clonal integration of MCPyV into the host genome, whereas VN-MCC is likely due to UV signature mutations[14,15]. We used small molecule screening to further characterize the distinct pathophysiologies of VP-MCC and VN-MCC, and to identify subtype-specific treatments. In our study, we describe disparate drug response signatures for VP-MCC and VN-MCC cell lines. Moreover, we identified multiple VP-MCC selective targets, implicating AURKs, histone epigenetic modification, and regulation of apoptosis

as critical processes in VP-MCC. Identification of apoptosis as a target and anti-apoptotic Bcl-2 family inhibitors as effective in MCC is consistent with a prior report that demonstrated in vitro and in vivo efficacy of the Bcl-2 family inhibitor navitoclax[48]. Having two of our three top screen targets validated in preclinical mouse models gives us confidence in our screening approach for VP-MCC.

In our screen, we identified 15 compounds that were VN-MCC selective; however, definitive conclusions about VN-MCC tumors based on our screening results must be made with caution. This is because the VN-MCC cell lines used may not be fully representative of native tumor biology[19], only positive hits on our finite initial screen were pursued for validation, and there were no shared target subcategories or compound classes among these compounds. Testing additional compounds for each of the targets identified as VN-MCC selective in our screen will help discover potential VN-MCC selective treatments.

Multiple small molecules and targets identified in our screen are in the same complex or pathway as AURKB, demonstrating the importance of this functional hub in MCC viability. Both our small molecule screen and prior reports identified the survivin inhibitor, sepantronium bromide, as a potent compound for inhibiting the viability of VP-MCC[32]. Moreover, sepantronium bromide slowed the growth of MKL-1 (VP-MCC) xenograft tumors[32]. Survivin, AURKB, INCENP and borealin comprise the CPC, which is responsible for cell cycle regulation and proper mitosis. The seemingly higher efficacy of AZD2811NP (relative to the reported effect of sepantronium bromide) on MCC xenografts may be due to the fact that AURKB is the only enzymatic component of the CPC. Nevertheless, small molecule inhibition of two members of the CPC (AURKB and survivin) independently inhibit VP-MCC xenograft growth in vivo, demonstrating the significance of the CPC complex to VP-MCC viability. In VP-MCC, a combination of AZD2811 and sepantronium bromide may show additive or synergistic effects and may help avoid resistance to AURKB inhibition.

Even though AURK inhibitors were more potent in VP-MCC cells than VN-MCC cells in vitro, AZD2811NP markedly slowed both VP-MCC and VN-MCC tumor growth in vivo. As discussed above, this may be due to the high plasma and tissue concentrations of AZD2811 achieved with AZD2811NP administration. It is also possible that the in vivo tumor microenvironment may be a second target for AZD2811. Taken together, our results suggest that AURK inhibitors may be a novel treatment option for patients with either VP-MCC or VN-MCC. Of note, a phase I clinical trial determined AZD2811NP was tolerated when given with granulocyte colony-stimulating factor (G-CSF)[49]. In addition, a combination of AZD2811NP and durvalumab, a PD-L1 inhibitor, is currently being tested in a phase II clinical trial (NCT04745689) for extensive stage small cell lung cancer, another neuroendocrine tumor. Results from this study may be useful in dose selection and estimating response rates in MCC patients for power calculations.

Inhibition of topoisomerase II with etoposide is part of standard chemotherapy treatment for patients with either VP-MCC or VN-MCC tumors[1,3]. The combination of cisplatin and etoposide effectively reduces tumor burden in most MCC patients, but resistance rapidly develops creating a need for effective combinations to delay or prevent this resistance. Topoisomerase II was identified and validated as a MCC target in our small molecule screens (Figs. 2A and 3A). Interestingly, topoisomerase II is required for correct localization of the CPC and functional activity of AURKB. Knockdown of topoisomerase II in *Drosophila* S2 cells does not change the level of AURKB protein, but H3Ser10 phosphorylation (a known AURKB substrate) is reduced to similar levels as AURKB knockdown alone[50]. As topoisomerase II and AURKs have functions that converge on the same pathway, it is possible that a combination of these targeted inhibitors may help prevent or delay resistance in MCC. Although multiple VP-selective targets were identified, including AURKB, there were no VN-selective target

subcategories discovered in our screen (relative to control or VP-MCC cells). The high mutational burden in VN-MCC tumors is associated with multiple, diverse candidate driver mutations[14,51]. This complexity and variability may result in a treatment-resistant phenotype or inconsistent responses to drug classes among VN-MCC tumors. Prior work has shown that, based on global gene expression, the VN-MCC cell lines are less representative of native MCC tumors than the VP-MCC cell lines[19]. The same study showed more variance among the transcriptomes of the three VN-MCC cell lines than the tightly clustered VP-MCC cell lines. This diversity may have contributed to the lack of drug classes that were consistently effective across the VN-MCC cells. The development of representative VN-MCC cell lines, patient-derived xenografts, or elucidating the driver mutations in individual VN-MCC human tumors may be necessary to predict the sensitivity of VN-MCC to small molecule drugs.

In contrast to the molecular heterogeneity of VN-MCC, VP-MCC are consistently driven by MCPyV oncogenes and VP-MCC cell lines are more representative of MCC tumors. All VP-MCC cell lines were extremely sensitive to killing by inhibitors of the CPC. One possible explanation for this is the dysregulation of the CPC by MCPyV oncogenes. In our studies, the knockdown of survivin or INCENP reduced the viability of MKL-2 (VP-MCC) but not MCC26 (VN-MCC). It has been reported that the knockdown of MCPyV large T antigen (MCPyV-LT) leads to a reduction in survivin mRNA and protein levels as well as induction of apoptosis in VP-MCC. In contrast, the knockdown of survivin failed to induce apoptosis in the VN-MCC cell-line, UISO[32]. By altering survivin, MCPyV-LT may cause aberrant localization or signaling by the CPC in VP-MCC (Supplementary Fig. S15). Such alterations could potentially explain why inhibition of AURKB, the enzymatic component of the CPC, potently reduces VP-MCC cell growth in vitro and shrinks tumors in vivo.

The small-T (ST) antigen of the MCPyV (MCPyV-ST) may also confer sensitivity to AURK inhibitors. MCPyV-ST leads to increased expression of the microtubule destabilizing protein, stathmin-1[39]. Pharmacological inhibition or knockdown of AURKB prevents hyperphosphorylation of stathmin-1 leading to microtubule depolymerization in *Xenopus* egg extracts[36]. Thus, MCPyV-ST may upregulate stathmin-1 thereby sensitizing the cell to mitotic collapse when AURKB is inhibited (Supplementary Fig. S15). Moreover, MCPyV-ST binds and inhibits PP2A phosphatases[52,53], which have been implicated in the regulation of both AURKB[54] and stathmin-1[55] (Supplementary Fig. S15). Thus, there are multiple potential mechanisms whereby MCPyV T antigens may enhance the activity of AURKB and the CPC in VP-MCC rendering them sensitive to inhibitors of the CPC.

AURKB was among the top targets identified for VP-MCC in both our small molecule and RNAi druggable genome screens. Integrating the small molecule and RNAi screening results led us to prioritize AURKB as a target and develop AZD2811NP, an AURKB selective inhibitor, as a potential treatment for MCC. Despite AZD2811NP being effective in treating both VP-MCC and VN-MCC xenograft tumors, the potency difference of AURKB inhibition observed in VP-MCC and VN-MCC cell lines suggests that tumor MCPyV status should be an analysis variable included in the design of trials translating AZD2811NP to the clinic. Similar applications of intersectional small molecule and RNAi screening may be helpful in discovering new targets, deconvolving the polypharmacology associated with small molecules, and aid in the discovery of novel treatments for other rare, understudied cancers.

## Methods

The research conducted complies with all relevant ethical regulations. Animal studies were approved by the Animal Care and Use Committee of the NCI-Frederick and animal care was provided in accordance with the procedures outlined in the "Guide for Care and Use of Laboratory Animals" (National Research Council; 2011; National Academies Press; Washington, D.C.). Frederick National Laboratory is accredited by AAALAC International and follows the Public Health Service Policy for the Care and Use of Laboratory Animals.

### Cell lines and culture

All MCC cell lines (WAGA[56], MKL-1[57], MKL-2[58], MCC13[59], MCC26[59], UISO[60]) and control cell lines (HACAT[61], HEK293T[62], CRL-7250[63], and NIH-3T3[64]) have been described previously. RPMI 1640, Dulbecco's modified Eagle's medium, phosphate-buffered saline (PBS), fetal bovine serum (FBS), penicillin-streptomycin solution, and trypsin were used for cell-culture and purchased from Thermo Fisher Scientific (Grand Island, NY). Accutase (MilliporeSigma, Burlington, MA, USA) was used for cell dissociation.

**Compound libraries.** The MIPE[18] and NPC[16] annotated drug libraries contained 1912 and 2816 compounds (after averaging duplicates), respectively.

**High-throughput small molecule screen.** A quantitative high-throughput screen (qHTS) was conducted in 1536-well white flat bottom plates (Corning) on a Kalypsys robotic system[65]. Cell lines were screened against NPC[16] and MIPE[18] libraries in dose-response (8 and 11 point respectively), measuring cell viability after 72 h. Briefly, cell lines were dissociated with trypsin or accutase (MKL-1 and MKL-2 only), passed through a 40 micron cell-strainer, and then plated with a Multidrop™ Combi Reagent Dispenser (Thermo Fisher Scientific, Grand Island, NY) into 1536 well plates and plated down at a starting density ranging from 50 cells/μL (HEK293T), 80 cells/μL (NIH-3T3), 100 cells/μL (MKL-1, MKL-2, MCC13, MCC26, HACAT, CRL-7250, and UISO), and 250 cells/μL (WAGA) in a final volume of 5 μL of media (MCC cells: RPMI 1640, Control cells: DMEM) supplemented with 10% FBS and 1X Pen/Strep. A 1536 pintool (Kalypsys, San Diego, CA, USA) was used to transfer 23 nL of compound in DMSO to the 1536-well assay plates. After 72 h incubation at 37°C and 5% $CO_2$, 2.5 μL of CellTiter-Glo (Promega, Madison, WI, USA) was dispensed into each well using a BiorapTR (Beckman Coulter, Fullerton, CA, USA). Plates were incubated at room temperature for 10 min, transferred to a ViewLux (PerkinElmer, Waltham, MA, USA), and the luminescence was recorded using an exposure time of 2 seconds. Relative luminescence units (RLUs) were normalized to in-plate controls (no cells as a positive control, DMSO as a negative control) and the normalized data was processed using NCATS in-house software[66]. Full qHTS data generated in this study are available at PubChem under accession number 1296009. A subset of 128 compounds from the initial qHTS were then retested in triplicate using an 11-point dose-response against six MCC cell lines (WAGA, MKL-1, MKL-2, MCC13, MCC26, and UISO) using identical conditions as above.

**qHTS data analysis and statistics.** The screening data was analyzed using software developed internally in NIH Chemical Genomics Center (http://tripod.nih.gov/curvefit/). Raw plate reads for each titration point were normalized plate-wise to intra-plate positive control (100% inhibition) and DMSO only wells (basal, 0%). The same controls were also used for the calculation of the Z'-factor index for each assay.

**Area under the dose response curve (AUC) analysis of high throughput-small molecule screen.** Area under the dose-response curve (AUC) was calculated for each dose-response curve using the trapezoidal formula:

$$\left( \frac{(log\,[max] - log\,[min]) * (response\,at\,[max] + response\,at\,[min] + 2 * (response\,at\,[intermediate]))}{2 * (\#\,of\,concentrations - 1)} \right)$$

AUCs < 0 were set to 0. AUC histograms for each cell line were plotted in Rstudio using ggplot. The 0.25 quantile AUC was calculated for each cell line based on AUCs for all compounds tested. The mean of these 0.25 quantile readouts (across all cell lines) was calculated (mean

= 449.6) and used as a threshold for compound-effectiveness (AUC < 450).

For analyses involving both NPC and MIPE libraries, if compounds were repeated within a library (e.g. multiple lots of the same compound) or between libraries, the average AUC/compound was used.

**Treemap summary analysis of high-throughput small molecule screen.** Treemap summary figure was created in Rstudio using treemap package. Compounds were categorized as "No effect" if AUC ≥ 450 in all 10 cell lines screened. Remaining compounds were further classified as "Pan-active" (AUC < 450 in all 10 cell lines screened), "Exclusive for control cells" (AUC < 450 in all 4 control cell lines screened and AUC ≥ 450 in all 6 MCC cell lines screened), or MCC-exclusive. MCC-exclusive compounds included those that were "Pan-MCC exclusive" (AUC < 450 in all 6 MCC cell lines screened and AUC ≥ 450 in all 4 control cell lines), "Exclusive for VP-MCC" (AUC < 450 in all 3 VP-MCC cell lines screened and AUC ≥ 450 in the 3 VN-MCC and 4 control cell lines screened) or "Exclusive for VN-MCC" (AUC < 450 in all 3 VN-MCC cell lines screened and AUC ≥ 450 in the 3 VP-MCC and 4 control cell lines screened).

**Volcano plots to quantify effectiveness of single agents.** Mean AUCs for each compound in the NPC/MIPE libraries were calculated within groups:
- MCC.avg: WAGA, MKL-1, MKL-2, MCC13, MCC26, UISO
- VP.MCC.avg: WAGA, MKL-1, MKL-2
- VN.MCC.avg: MCC13, MCC26, UISO
- Control.avg: HACAT, HEK293T, CRL-7250, NIH-3T3

and the difference in mean AUC between groups was calculated:
- Delta.MCC.control=Control.avg-MCC.avg
- Delta.MCVpos.control=Control.avg-VP.MCC.avg
- Delta.MCVneg.control= Control.avg-VN.MCC.avg

A two-sided t-test (using defaults in the t.test stats package in Rstudio) was used to calculate the p-value for differences in mean AUC between MCC cell lines (either all 6 MCC, VP-MCC, or VN-MCC) and control cells lines. The -log base 10 of this p-value was used to create plots in Rstudio using ggplot2 and ggrepel packages. Statistical significance was set at a delta > 50, p-value < 0.05, and the AUC for all MCC cell lines in comparison (either all MCC, VP.MCC, or VN.MCC) of < 450.

**Comparison of primary NPC/MIPE screen and follow-up validation small molecule screens.** AUC results were subsetted from the primary high-throughput small molecule screen for all compounds that were in the follow-up screen. Compounds were excluded if different concentrations were used in the two screens. Data was processed using base and reshape2 packages, correlation coefficients were calculated using cor.test command in stats package, and plots were created with ggplot2 (Rstudio). PCA analysis was performed as described below.

**Principal component analysis (PCA) of cell lines.** PCAs were performed using prcomp in stats package in Rstudio using default parameters (except scale=TRUE). PCA was plotted using scatterplot3d or plot3D and rotating in open3d in Rstudio.

**Heatmap analysis.** Heatmaps were rendered in Rstudio using pheatmap and RColorBrewer packages.

**Target subcategory and compound class annotation and analysis.** All compounds in the MIPE library were annotated based on two hierarchical categories: target subcategory (n = 156; target subcategories only included if > 2 compounds/subcategory) and compound class (n = 56; only included if > 3 compounds/class)[33]. The compound class is a broader category that groups compounds based on their general pharmacological or therapeutic function. Within each compound class, the target subcategory provides a more specific classification based on the molecular target or mechanism of action of the compounds. This dual classification system helps in understanding both the overarching therapeutic application and the precise biological target of each compound. AUC was averaged within each cell line for all compounds from the same target subcategory or compound class. This aggregated AUC (by cell line) for each target subcategory or compound class was then averaged across:
- MCC cell lines (WAGA, MKL-1, MKL-2, MCC13, MCC26, UISO)
- VP-MCC (WAGA, MKL-1, MKL-2)
- VN-MCC (MCC13, MCC26, UISO)
- Control cell lines (HACAT, HEK293T, CRL-7250, NIH-3T3)

In Fig. 1E, the average MCC (MCC.avg) and control (Control.avg) AUCs for each target subcategory were compared and statistical significance (shown as orange bubbles) was set by 3 parameters: 1) Control.avg-MCC.avg > 40, 2) p-value < 0.05, and 3) AUC < 450 for all MCC cell lines. For Fig. 2A, the average VP-MCC AUC was subtracted from the average VN-MCC AUC for each target subcategory and these differences were ranked. Positive and negative differences correspond to VP-MCC or VN-MCC selective reduction in viability, respectively. Statistically significant (purple bars) differences in viral selectivity for the target subcategory were determined by 3 parameters: 1) a difference in average VP-MCC vs. VN-MCC viability > 40, 2) p-value < 0.1, and 3) AUC < 450 for either all VP-MCC or all VN-MCC cell lines (depending on viral selectivity). For Fig. 2B, the average VP-MCC (VP-MCC.avg) and VN-MCC (VN-MCC.avg) AUCs for each compound class were compared and statistical significance (shown as purple bubbles) was set by 3 parameters: 1) |(VN-MCC.avg)-(VP-MCC.avg)| > 30, 2) p-value < 0.1, and 3) AUC < 450 for either all VP-MCC or all VN-MCC cell lines (depending on viral selectivity). For Supplementary Fig. S4A, B, VP-MCC.avg (A) or VN-MCC.avg (B) was compared to control (Control.avg) AUCs for each target subcategory and statistical significance (shown as purple bubbles) was set by 3 parameters: 1) Control.avg-VP-MCC.avg > 40 (A) or Control.avg-VN-MCC.avg > 40 (B), 2) p-value < 0.05, and 3) AUC < 450 for either all VP-MCC (A) or all VN-MCC (B) cell lines. All p-values were calculated in Rstudio using t.test in stats package with default parameters, plotting used ggplot2 and ggrepel.

**Radar chart analysis.** The average AUC was calculated for each AURK inhibitor (5 total) in each MCC cell line. The cell-line specific averages were then averaged across VP-MCC and VN-MCC cell lines for each AURK inhibitor. The mean AUCs were plotted in Rstudio using radarchart in the fmsb package using a value range between 200 and 500.

**Dose response curve analysis and IC$_{50}$ calculations.** Dose-response curves were fit by nonlinear regression using a three or four-parameter logistic equation and IC$_{50}$ was calculated using GraphPad Prism (La Jolla CA, USA).

**The dependency map (DepMap) data analysis.** The depMap database[34] IC$_{50}$ results were exported on 5/27/2024 for AZD2811 and combined with IC$_{50}$ of AZD2811 from our small molecule screen calculated by GraphPad Prism. Compounds were ranked by IC$_{50}$ and graphed using GraphPad Prism (La Jolla CA, USA).

**Genomics of drug sensitivity in Cancer (GDSC) data analysis.** The GDSC database[35] IC$_{50}$ results were exported on 05/21/2018 for tozasertib and GSK-1070916A and combined with IC$_{50}$ estimates (calculations performed in TIBCO Spotfire: Palo Alto, CA, USA) for these compounds from our small molecule screen. Following the graphing format in the GDSC database, compounds were ranked by IC$_{50}$ and plotted in Rstudio using ggplot2, ggrepel, and scales packages.

**RNAi Druggable genome screen.** Assay development prior to screening demonstrated the transfectability of the cell lines as assessed by transfection efficiency using appropriate controls. MCC26 and MKL-2 were selected because they demonstrated higher transfection efficiency than other MCC cell lines. A non-targeting control (Silencer Select Negative Control No. 2 siRNA) and a positive control (AllStars Cell Death Control siRNA) were used as controls. Lipofectamine RNAiMax (Thermo Fisher Scientific, Grand Island, NY) was obtained as the transfection reagent.

The Silencer Select Human Druggable siRNA library (Thermo Fisher Scientific, Grand Island, NY), which targets 10,418 human genes, was used for primary screening. Three siRNAs per gene were arrayed in 384 well plates, one siRNA per well. For each well, 2 µL of siRNA was complexed with 0.15 µL RNAiMax Lipofectamine transfection reagent (Thermo Fisher Scientific, Grand Island, NY) in 20 µL of serum-free RPMI 1640 (Thermo Fisher Scientific, Grand Island, NY) media for 45 minutes at ambient room temperature. 500 MCC26 cells and 750 MLK-2 cells in 20 µL of RMPI 1640 and 20% fetal bovine serum (Millipore Sigma, Burlington, MA, USA) were plated and kept at ambient room temperature for five minutes before incubation at 37°C/5% CO2 for 48 h. Cell viability was measured by CellTiter-Glo assay following the manufacturer's instructions (G7572; Promega, Madison, WI, USA) 96 h after siRNA transfection. Assay plates were measured using BMG Pherastar plate reader

Full data for the RNAi druggable genome screen are available in Supplementary Dataset 3.

**Assessment of cell viability (WST-1).** For non-screening cell viability measurements (Fig. 4E) the WST-1 Cell Proliferation Assay Kit (Clontech, Mountain View, CA, USA) was used. After 3 days of treatment, WST-1 solution was added to cell suspension in a 96-well plate according to the manufacturer's protocol. The formazan product was measured by absorbance on a plate reader (Victor X3, Perkin-Elmer, Waltham, MA, USA).

**siRNA to AURKB included in follow-up knockdown studies.** AURKB siRNA (Catalog: 4390825; Assay ID: s17612, sequence: UCGUCAAGGUGGACCUAAAtt) (Thermo Fisher Scientific, Grand Island, NY) was selected for follow-up study.

**siRNA to INCENP.** Three different siRNAs (sequences: GCUUGUACCUCAUAUCAGAtt, AGAUCAACCCAGAUAACUAtt, and AGUCCUUUAUUAAGCGCAAtt) targeting INCENP as well as negative non-targeting siRNA control were purchased from Thermo Fisher Scientific (Grand Island, NY). MCC cells (MKL-2 or MCC26) were reverse transfected using Lipofectamine RNAi/MAX (Thermo Fisher Scientific, Grand Island, NY) and viability was measured after a 96-h incubation using CellTiter-Glo (Promega, Madison, WI, USA)

**Nucleofection of AURKB siRNA (VP-MCC cell lines).** In follow-up knockdown studies (Fig. 4C-E), VP-MCC cells (WAGA or MKL-1) were pelleted, counted, and combined with AURKB siRNA (Catalog#: 4390825; Assay ID: s17612; Thermo Fisher Scientific, Grand Island, NY) in nucleofector solution (WAGA: SG Cell Line 4D-Nucleofector® X Kit or MKL-1: SF Cell Line 4D-Nucleofector® X Kit; Lonza, Walkersville, MD, USA) and shocked (shock pattern WAGA: CM-138, MKL-1: CM-135) according to the manufacturer's protocol. Shock without AURKB siRNA was used as a negative control.

**siRNA knockdown of AURKB in VN-MCC using Lipofectamine RNAi/MAX.** In follow-up knockdown studies (Fig. 4C–E), VN-MCC cells (MCC13 and UISO) were reverse transfected using Lipofectamine RNAi/MAX (catalog#: 13778150; Thermo Fisher Scientific, Grand Island, NY) with AURKB siRNA (Catalog#: 4390825; Assay ID: s17612; Thermo Fisher Scientific, Grand Island, NY) according to the manufacturer's protocol (adjusted for plate/well volume).

**Nuclear Extraction.** Nuclear extraction of MCC cells was performed 48 h after transfection with AURKB siRNA using a Nuclear Extraction Kit (ab113474, abcam, Cambridge, MA, USA) according to the manufacturer's protocol including sonication after vortexing the extract.

**Harvesting of whole cell lysates.** Cells were washed two-times in ice-cold PBS and then pellets were incubated on ice in lysis buffer (Cell Signaling Technology, Danvers, MA, USA) supplemented with protease inhibitor cocktail (cOmplete™, Mini Protease Inhibitor Cocktail, Millipore Sigma, Burlington, MA, USA) and phosphatase inhibitors (PhosSTOP- phosphatase inhibitor tablets, Millipore Sigma, Burlington, MA, USA). Lysed cells were spun down at 14,000 RCF at 4°C for 10 minutes.

**Drug treatment and histone extraction.** VP-MCC (WAGA, MKL-1, and MKL-2) and VN-MCC (MCC13 and MCC26) cells were plated at a concentration of $2.5 \times 10^5$ cells/mL (RPMI 1640 supplemented with 10% FBS, 100 U/mL penicillin and 0.1 mg/mL streptomycin) and then treated with AZD2811 (ordered as AZD1152-HQPA from MilliporeSigma, Burlington, MA, USA or Selleckchem, Houston, TX, USA) (3 nM or 30 nM) or DMSO-control alone.

After a 24-h incubation (72 h for MCC26), MCC cells were washed with 5 mM sodium butyrate and then resuspended in triton extraction buffer (1X PBS, 0.5% Triton X-100, phosphatase inhibitor tablet, and protease inhibitor cocktail), at a concentration of $4 \times 10^7$ cells per ml. Cells were rocked on ice for 10 minutes and then centrifuged at 6500 x $g$ for 10 minutes at 4°C. Cell pellets were resuspended in 0.2 N HCL and kept at 4°C overnight before centrifugation at 6500 x $g$ for 10 minutes at 4°C. The supernatant containing histone proteins was used for western blotting.

**Western Blotting.** A Bradford assay (Bio-Rad Protein Assay Dye Reagent Concentrate, Bio-Rad, Hercules, California, USA) was used to measure protein concentration. Protein (histones: 2 µg protein, nuclear extracts and whole cell lysates: 20-35 µg) was prepared in 2X tris glycine SDS sample buffer (Thermo Fisher Scientific, Grand Island, NY) containing 4% β-mercaptoethanol and separated on 4-20% SDS-PAGE gradient gels (Thermo Fisher Scientific, Grand Island, NY) and transferred to PVDF membranes (Thermo Fisher Scientific, Grand Island, NY). Membranes were blocked for 1 h with blocking buffer (Protein Simple, San Jose, California, USA) and then probed with primary antibodies against AURKB (ab45145, 1:500, abcam, Cambridge, MA, USA), β-actin (sc-47778, 1:200, Santa Cruz Biotechnology, Dallas, Texas, USA), Stathmin-1 (ab52630, 1:1000, abcam, Cambridge, MA, USA), phospho-Histone H3 (Ser10) (06-570, 1:150, Millipore Sigma, Burlington, MA, USA), Histone H3 (3638, 1:300, Cell Signaling Technology, Danvers, MA, USA), PARP (9532, 1:1000, Cell Signaling, Danvers, MA, USA), or MCPyV large T (CM2B4, 1:200, Santa Cruz Biotechnology, Dallas, Texas, USA). After washing membranes according to the manufacturer's protocol (ProteinSimple, San Jose, California, USA), proteins were visualized using fluorescent secondary antibodies (1:1,000, ProteinSimple, San Jose, California, USA) and Alphaview software (ProteinSimple, San Jose, California, USA) was used to quantify molecular weights and intensity of protein bands.

**Flow cytometry analysis.** MCC cells (WAGA, MKL-1, MKL-2, MCC13, MCC26) were treated with 30 nM of AZD2811 or DMSO control. Following a 24-h incubation (24 and 72 h for MCC26), cells were washed in ice-cold PBS, and fixed in 75% ethanol for at least 24 h. Following fixation, cells were stained with 50 ug/mL of propidium iodide (Biolegend, San Diego, CA, USA) containing 100 ug/mL RNAse A (MilliporeSigma, Burlington, MA, USA) per million cells for 3 - 4 h at 4°C. DNA content was measured on BD FACSCanto II (BC Biosciences, Franklin Lakes, NJ, USA) or Invitrogen Attune NxT Flow Cytometer (Life Technologies, Carlsbad, CA, USA) and analyzed with FlowJo (Ashland, OR, USA) using auto-gating and median FSC-H.

**Immunohistochemistry.** Paraffin-embedded MCC13 (VN-MCC) tissue sections were deparaffinized, rehydrated, and treated with Bloxall solution (SP-6000, Vector Laboratories, Burlingame, CA, USA) at room temperature to block endogenous peroxidase activity, followed by heat-mediate antigen retrieval using sodium citrate buffer (Bio SB, Santa Barbara, CA, USA). The MCC13 tissue sections were then blocked with goat serum (Vector Laboratories, Burlingame, CA, USA) for 30 minutes at room temperature before incubation with primary antibodies at 4°C overnight. The following day, the samples were incubated with peroxidase-labeled secondary antibody (Vector Laboratories, Burlingame, CA, USA) for 30 minutes at room temperature, followed by diaminobenzidine (DAB) (Vector Laboratories, Burlingame, CA, USA) reaction and counterstaining with hematoxylin (Millipore Sigma, Burlington, MA, USA). The MCC13 xenograft sections were stained with primary antibodies against Ki-67 (ab16667, 1:200, Abcam, Cambridge, MA, USA), phospho-Histone H3 (Ser10) (06-570, 1:500, Millipore Sigma, Burlington, MA, USA), or cleaved caspase-3 (9661, 1:400, Cell Signaling Technology, Danvers, MA, USA). Rabbit Elite Vecstatin ABC IHC kit (PK-6101, Vector Laboratories, Burlingame, CA, USA) and Immpact DAB Peroxidase Substrate kit (SK-4105, Vector Laboratories, Burlingame, CA, USA) were used for immunohistochemistry. Positive cells were blindly quantified from 4–6 20x images per tissue section (n = 3 animals per treatment group).

**Immunofluorescence.** VP-MCC cell lines WAGA and MKL-1 cells were immobilized on glass microscope slides using Shandon Cytospin 2 centrifuge (MKL-1 were accutased for 5 min to obtain single cell suspension before cytospin). VN-MCC cell lines MCC13 and MCC26 were seeded on glass coverslips 24 h before staining. For immunofluorescence staining, all MCC cell lines were fixed with paraformaldehyde 4%, permeabilized by Triton X-100. After permeabilization and washing with 5% normal goat serum/PBS, a primary antibody against pericentrin (ab4448, 1:2000, Abcam, Cambridge, MA, USA) was applied to the cells for 1 h at room temperature. After washing, cells were incubated with fluorescent secondary antibodies for 1 h at room temperature. Cells were mounted using VEC-TASHIELD Antifade Mounting Medium with DAPI (H-1200-10, Vector Laboratories, Burlingame, CA, USA). Positive cell quantification and cell size measurements were performed blindly from 4–6 40x images per slide (n = 3 replicates per treatment condition).

**Caspase-Glo assay.** Apoptotic effects of AZD2811 on MCC cell lines were measured using Caspase-Glo 3/7 reagent from Promega (Madison, WI, USA). Cells were plated in quadruplicate at a concentration of $25 \times 10^4$ cells/mL for VP-MCC (WAGA and MKL-1) and $10 \times 10^4$ cell/mL for VN-MCC (MCC13 and MCC26). Following overnight incubation, cells in Falcon® 96-well White/Clear Flat Bottom plates (Corning Inc., Corning, NY, USA) were treated with increasing concentrations of AZD2811 for 72 h. At the time of apoptosis measurement, plates were allowed to reach room temperature for 30 min. 100 μL of Caspase-Glo 3/7 assay reagent was added to each well, plates were covered, placed on a shaker for ~1 h and luminescence was measured (Victor X3 Multilabel Reader, Beckman Coulter, Fullerton, CA, USA). Caspase activity was normalized to DMSO control and calculated by averaging 4 replicates.

**Biochemical profiling of AURK inhibitors.** AURK inhibitors were tested at 10-doses starting at 10 μM with 3-fold serial dilutions. Staurosporine (control compound) was also tested at 10-doses but starting at 20 uM with 4-fold serial dilutions. All reactions were carried out at Km ATP. Assay was performed by Reaction Biology (Malvern, PA, USA).

**AURKB transcript expression.** The Gene Expression Omnibus (GEO) accession number GSE50451 data were analyzed[44]. Probe IDs corresponding to AURKA (204092_s_at, 208079_s_at, 208080_at), AURKB (209464_at), and AURKC (211107_s_at) were identified for Affymetrix

Human Genome U133A 2.0 Array (GPL571) and the GEO2R tool was used to identify transcript expression levels for MCC human tumor samples included in the study. The average transcript level for AURKB (n = 3 probe sets) was plotted.

**Mouse xenograft studies.** Frederick National Laboratory is accredited by AAALAC International and follows the Public Health Service Policy for the Care and Use of Laboratory Animals. Animal studies were approved by the Animal Care and Use Committee of the NCI-Frederick and animal care was provided in accordance with the procedures outlined in the "Guide for Care and Use of Laboratory Animals" (National Research Council; 2011; National Academies Press; Washington, D.C.).

Female, athymic Nu/Nu mice, 4 to 6 weeks old were obtained from Charles River Laboratories and housed at Frederick National Laboratory for Cancer Research. Female mice were selected for the study due to considerations of housing costs and available space, as they can be housed together. Mice were subcutaneously injected with 10 million MKL-1 cells (25 G needle; Med-Plus, Inc., Edison, NJ) or 5 million MCC13 cells (23 G needle; Med-Plus, Inc., Edison, NJ) in 50% Matrigel (Corning, Corning, NY)/50% PBS in a total volume of 100 microliters (MKL-1) or 300 microliters (MCC13). Once the average tumor volume reached ~100 mm³, animals were randomized into treatment groups using the StudyLog software randomization tool (Pacifica, CA, USA) and were treated with 25 mg/kg (10 mL/kg of bodyweight) of AZD2811NP (Cambridge, AstraZeneca, UK) intravenously in 0.9% saline (or placebo nanoparticles) two times/week (Monday and Wednesday) for four weeks for efficacy studies and 10 days for mechanistic studies. Tumor size and body weight were measured on average two times or three times per week, respectively. After four weeks of treatment, we continued to monitor tumor size and body weight twice/week until all animals were euthanized (MKL-1: 242 days, MCC13: 114). Study endpoints were body weight loss exceeding 20% or tumor size exceeding 20 mm in diameter.

**In vivo Pharmacokinetics study.** Female nude mice (n = 33) were given a single IV administration of AZD2811NP (25 mg/kg). Blood (~200 μL) and liver (~0.5 g) samples were collected at the following time points: 0, 0.168, 3, 7, 24, 48, 72, 120, 168, 336, and 504 h (n = 3 animals/time-point). UPLC-MS/MS method was developed to quantify AZD2811 in plasma and liver homogenate. PK parameters (e.g. AUC, Cl, $V_{ss}$, $C_{max}$, $T_{max}$, $t_{1/2}$) were calculated using WinNonLin software (Randor, PA, USA).

**Statistical analysis and reproducibility.** Experimental design: we designed automated high-throughput small molecule (n = 1) and RNAi druggable genome (n = 1) screens for VP-MCC and VN-MCC cells in culture. A follow-up small molecule validation screen (n = 3) was conducted on a subset of compounds from the primary small molecule screen. Mechanistic studies were performed with at least three repetitions. 10 mice were used for each in vivo efficacy study and 3 mice for each mechanistic study arm. Mice with established xenograft tumors were randomized into vehicle and treatment arms.

Statistical analysis: dose-response curves were fit by nonlinear regression using either a three or four-parameter logistic equation and $IC_{50}$ values were calculated in GraphPad Prism (La Jolla CA, USA). P-values were calculated for target subcategory and compound class bubble plots/barcharts in Rstudio without correction. Dendrogram clustering was performed in Rstudio (Fig. 1D, Supplementary Figs. S3A and S5A) using default parameters in the Pheatmap package. PCA (Fig. 1C and Supplementary Fig. S5B) was performed in Rstudio using prcomp package. One-sample t-test was used when comparing to a set value of 100 while a two-sample t-test (paired when appropriate) was used when comparing between two groups. For in vivo studies, tumor growth was plotted over time, and linear regression was

used to compare rates of growth in GraphPad Prism (La Jolla CA, USA). Kaplan-Meier curves were constructed and analyzed using Log-rank (Mantel-Cox) test in GraphPad Prism (La Jolla CA, USA).

Experiments were repeated multiple times under the same conditions to ensure reproducibility. All results were consistent across independent trials. No statistical method was used to predetermine the sample size. No primary data points were excluded except when they failed the required criteria of the data analysis being applied as described in the methods section. Randomization was implemented for the in vivo experiment. For experiments requiring quantification, blinding was implemented to reduce bias.

### Reporting summary

Further information on research design is available in the Nature Portfolio Reporting Summary linked to this article.

### Data availability

Full qHTS data generated by this study are available at PubChem under accession number 1296009. Arrayed RNAi screen data are available in Supplementary Dataset 3. The results from the experiments performed in this study are provided as a Source Data file.

### Code availability

No unique or custom computer code or algorithm was used in this work.

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

## Acknowledgements

We thank Dr. Jay-Hyun Jo for the scientific discussion and bioinformatics advice. We thank Dr. Simone Difilippantonio, Chelsea Sanders, and the rest of the Laboratory Animal Sciences Program at the Frederick National Laboratory for Cancer Research (NIH) for their dedication to animal studies. We thank AstraZeneca for generously providing AZD2811NP and control nanoparticles. We thank Dr. Jürgen Becker for providing WAGA, MKL-1, MKL-2, and UISO cells. We thank Dr. Patrick Moore for providing MCC13 and MCC26 cells. We thank Dr. Javed Khan's lab for providing HACAT, HEK293T, CRL-7250, and NIH-3T3 cell lines. We thank the Developmental Therapeutics Program at the NIH for supplying us with compounds for in vivo studies. We thank the NCI Flow Cytometry Core Facility in the Laboratory of Genome Integrity at the NIH. This work was supported by the National Institutes of Health (NIH) Intramural Research Program, the National Cancer Institute, the National Center for Advancing Translational Sciences, and the National Institute of Arthritis and Musculoskeletal and Skin Diseases (ZIA AR041222 to IB). Additional work was funded by the National Cancer Institute, NIH under Contract No. HHSN261200800001E. The opinions expressed are those of the authors and do not necessarily reflect the views or policies of the Department of Health and Human Services, nor does mention of trade names, commercial products, or organizations imply endorsement by the U.S. Government.

## Author contributions

T.G. experimental design, oversaw and ran experiments, data analysis, and integration, and wrote the manuscript. K.G. experimental design oversaw and ran experiments, data analysis, and integration, and reviewed and edited the manuscript. D.U. experimental design and ran high-throughput small molecule screening. A.C. and R.G. experimental design and ran molecular biology experiments. B.G. data analysis and

small molecule annotation. T.L. optimization and small molecule screening. O.L. and M.S. high-throughput small molecule screening data normalization and analysis. S.C., K. C., and M.L.N experimental design and data analysis for RNAi druggable genome screen. J.B. bioinformatics and data analysis. Y.X., L.M., J. E. J., T.S.R., and N.T.H. data analysis. S.D. experimental design for animal studies. A.Q.W. experimental design for pharmacokinetic animal studies and editing of the manuscript. M.H. and I.B. oversaw experimental design and manuscript writing. All authors contributed to the review and editing of the manuscript.

## Funding

## Competing interests

The authors declare no competing interests. US Provisional Patent Application No. 62/657,398 was submitted based, in part, on data reported here.
