## [Transparent Peer Review file · Nature Communications]

High-throughput screening identifies Aurora kinase B as a critical therapeutic target for Merkel cell carcinoma

Corresponding Author: Dr Isaac Brownell

Version 0:

Reviewer comments:

Reviewer #1

(Remarks to the Author)

Merkel cell carcinoma (MCC) is a deadly cancer with a highly unmet medical need. Here, the authors ran a screen of ~4000 drugs to assess of virus positive and virus negative MCC cell lines at two different concentrations of the compounds. In parallel, a RNAi screen was run; both modalities (re: small molecule and siRNA) identified Aurora Kinase B as a critical therapeutic target. This reviewer applauds the approach. Overall the paper is very thorough, well-executed, with impressive data analysis and provides sufficient explanation and follow-up studies of the screening hit(s). The major drawback is lack of novelty, as explained in more detail below. As a point of comparison, note that a NatComm paper (Paulson et al) was just published this week regarding class I HLA and resistance to immunotherapy in MCC, which by contrast to the present work, was a finding quite unexpected and novel (and worthy of publication in this journal).

Specific Comments:

1. Background

- a. Paragraph # 3 (starting with "Aurora kinase B...") seems to be out of place, giving background information about the hit before describing the screen. It may be better positioned in the end of the "Background" section.
- b. Given the caliber of the journal, paragraph # 3 attests to the fact that the gene hit is not really novel (re: ref 16 and the tie-in to survivin pathway)
- c. End of paragraph # 4 –worth stating that the cell lines are human.

2. Results

- a. General note: All figures that have relevant information in supplementary material should have a reference to relevant supplementary figures in the legends.
- b. Sub-Section: Identification of VP-MCC and VN-MCC-selective drug targets and compounds
 - i. Paragraph # 1 indicates that there were 99 VN-MCC selective hits, yet paragraph # 2 states that there were 156 initial target subcategories. How is this possible? Were a lot of compounds non-specific? Also, here it would be better to refer to compounds as drugs, not as "targets" as no compound will have absolute specificity for a given target.
 - c. Sub-Section: Follow-up small molecule screens support AURKs as a high priority target for VP-MMC
 - i. Line # 3: were the listed AURK inhibitors included in the primary screen or are these additional compounds used for validation?
 - d. Sub-section AURK signaling partners in the CPC also maintain VP-MCC viability. Why are we not shown the individual knockdown data for the three proteins seen in the Chromosomal Passenger Complex (CPC) of which AURKB is a part of, that is, survivin, INCENP, and Borealin? It would help to substantiate the author's claims. Note: the tie-in to stathmin-1 and gastric/nasopharyngeal cancers is very perceptive
 - e. Sub-section: RNAi screening independently identifies aurora kinase B as a MCC therapeutic target
 - i. Why were MKL-2 and MCC-26 lines chosen for the RNAi screen?
 - ii. If both VP-MCC and N-MCC provide medial challenges (though in different percentages of patient population, why do the authors continuously select hits specific to VP lines that are necessarily excluded from VN?
 - f. Section: Inhibition of aurora kinase B reduces tumor burden and extends survival in preclinical mouse models of MCC
 - i. Paragraphs # 2 and 3: The authors show that that animals injected with MKL-1 cells respond better to AZD2811NP than animals that were injected with MCC-13. This is demonstrated by their longer life span (>200 days) compared to MCC-13 treated with AZD2811NP (~120 days). However, if you compare the placebo-treated groups, mice injected with MCC-13 die sooner than MKL-1-injected mice. Are MCC-13 tumors more aggressive? If so, the fold increase in survival

(AZD2811NP/placebo mice) is roughly 2X in tumors from both cell lines.

3. Discussion

- a. Paragraph # 4: Fig. 2A shows five drug target categories that ranked higher than AURK. Why did the authors decide to focus on this pathway?
- b. Paragraph # 4: Given the known efficacy of sepantonium bromide, have the authors tried to combine this treatment with AZD2811NP? Would they expect an additive/synergistic effect?
- c. 2nd to last paragraph: If AZD2811NP is currently being tested for small cell lung cancer, would these findings be considered sufficiently novel?

4. Figure 2

- a. Why were VN-MCC targets actively avoided?
- b. The divergence of selected categories from the diagonal line is quite weak.

5. Figure 3

- a. Fig. 3c -Compared to other compounds shown in Fig. S6, Tozatinib shows a relatively weak response in VP-MCC cell lines. Why was it chosen for the main figure? If it is included in order to show the range of responses, it should be stated in the text.

6. Figure 4

- a. One of the reasons the authors chose AURKB, rather than other isoforms, was due to the difference in efficacy of its knock-down on VP vs. VN cell line viability. However, based on Fig. 4E, the response in MCC-26 cell line is not representative of other VN cell lines. Why did the authors choose MCC_26 cell line to determine their hits?

7. Figure 3D and Figure S6c

- a. Fig. 3D contains information from 420 cell lines from GDSC, whereas Fig. S6c contains 950 cell lines. Why is there such a large difference?

Reviewer #2

(Remarks to the Author)

In this manuscript by Gelb et al, they screened nearly 4000 drugs to search for potential chemotherapeutic drugs for Merkel cell carcinoma (MCC). Their AUC analyses combined with principal component analysis, hierarchical clustering, and bubble plot analysis identified that small molecules targeting AURK, BCL-xL, FLT, HDAC, and topo II selectively suppress the viability of Merkel cell polyomavirus (MCPyV)-positive(VP)-MCC. RNAi kinome screening further confirmed that AURKB inhibition is the most effective in selectively reducing the viability of VP-MCC cells. They also showed the importance of AURKB downstream signaling for VP-MCC cell survival by knocking down AURKB signaling partners, including Survivin and INCENP as well as AURKB substrate Stathmin-1, which also reduces MCPyV(+) MCC cell viability. Consistent with the inhibition of mitotic kinase, reduction in MCC cell viability by AURKB specific inhibitor AZD2811 was associated with cell cycle arrest in G2M. Finally, by using a xenograft mouse model, the authors showed that nanoparticle-formulated AZD2811NP inhibits the xenograft growth of both MKL-1 and MCC-13.

MCC is the 2nd most common cause of skin cancer death, next to melanoma. Prior to the discovery of MCPyV within MCC, cancer cell signaling pathways that could be targeted for MCC chemotherapy had been unknown. This manuscript demonstrated that the AURKB inhibitor selectively inhibits VP-MCC by unbiased screening method. If the AURK pathway is activated by MCPyV and is critical in virus-induced MCC genesis, AURKB is the rational chemotherapeutic target, and the AURKB inhibitor may be promising for future clinical trials.

General comments:

Overall, the manuscript is well-written, and the conclusion is supported by the presented results. One missing component, however, is the mechanistic insight into the selective inhibition of MCPyV(+) MCC cancers. Additionally, the cellular mechanism that decreased viability by the AURKB inhibitor is unclear.

Specific comments:

- In the paper, authors provide one example that viral activation of Stathmin-1 activation may explain AURKB-dependence of MCPyV(+) MCC by showing higher Stathmin-1 expression in VP-MCC. Does Merkel cell virus oncogene such as viral large T or small T expression affect AURKB expression? Is AURKB highly activated in VP-MCC?
- Authors focus on decreased viability induced by AURKB inhibition in WST assays. Is decreased viability due to mitotic arrest or cell death induced by apoptosis or mitotic catastrophe? While results in Fig 5C suggest G2/M cell cycle arrest is due to decreased viability, authors needs to clarify whether drug treatment is associated with cell death both in vivo and in vitro.
- VP-MCC cell lines used in each assays are not consistent. In the AURKB knockdown analysis, WaGa and MKL-1 are used for viability evaluation (Fig. 4E), while WaGa and MKL-2 are used in AZD2811 drug assays and MKL-1 is used in xenograft assays. At the very least, MKL-1 needs to be included in all the assays.
- Viral T antigen expression in VP-MCC has been shown to be required for MCC cell growth. Does AZD2811 affect viral T antigen expression?

Reviewer #3

(Remarks to the Author)

This manuscript, by Gelb, Brownell, and colleagues, describes screens of known agents and RNAi targets that converge on the identification of the potential utility of Aurora B kinase inhibitors as therapeutic options in Merkel cell carcinoma. Since the small agent most fully profiled (AZD2811-NP) is in use clinically, this work offers a clear path toward clinical translation –

a significant impact. Overall the work is quite complete and follows through from in vitro screening to in vivo results. Further, the paper is generally well written, with a bare minimum of typographical and grammatical errors. This work is likely suitable for publication in Nature Communications, but there are a few scientific questions and comments in this reviewer's mind that need to be addressed:

1. In the abstract, as well as occasionally in the text, the authors state that this work identifies Aurora B as a novel target. This is a bit of an overstatement, since the kinase is well established as a cancer drug discovery target, and clinical stage agents exist. The statement should always be qualified with "in Merkel cell carcinoma".
2. The 156 target subcategories could use some additional brief explanation, especially given the lack of associations with the inhibition phenotypes. How granular are the target designations (e.g. all kinases inhibitors in one class vs sufficient target detail to separate out different kinases or branches of the kinome, etc)? In addition to this, most agents, likely including many of those used in this study, have multiple molecular targets. How annotated are the target classes for each compound? It is entirely conceivable that one compound may fit into more than one subcategory; a slightly different grouping with more such detail might significantly change the lack of associations (or lack of associations) observed.
3. In the results section describing the follow-up screening with small molecules (Fig 3 and S6), it is stated that modest effects were observed at high concentrations with certain Aurora inhibitors. However, both Figure 3C and S6A, some of the small molecules appear to show full inhibition of at least some of the VN-MCC lines. While the potencies are less than in the VP-MCC lines on average, the full efficacy observed at higher concentrations for many of them does not seem modest.
4. The most confusing aspect of the work is the disconnect between some of the results. For example, in the small molecule target subcategories work (Fig 2 and attendant discussion), Aurora inhibitors are shown to be selective for VP-MCC vs VN-MCC. This notion is backed up by cellular work with AURKB inhibitor molecules (Fig 3 and discussion), and also by a portion of the RNAi results, in which AURKB knockdown is ineffective in at least one VN-MCC line. However, the extended sampling of RNAi against AURK in additional MCC lines points toward the MCC-26 line being a bit of an outlier, rather than representative of VN-MCC as a whole. This seems to indicate that AURKB inhibition is not as selective for VP- vs VN-MCC as claimed earlier in the text.
5. Related to point 4 above, AZD2811 is indeed less effective in inhibition of proliferation in the VN-MCC cell lines, including MCC-13. However, AURKB knockdown with RNAi is effective in MCC-13, and AZD2811 is shown to work in vivo in an MCC-13 derived xenograft. The authors offer that changes in cellular size are related to the differing effectiveness of the molecule, but this explanation seems not wholly sufficient in light of the in vivo activity of AZD2811 in MCC-13 (which was resistant to changes in cellular size). Are there other underlying reasons for the differences? A few additional possibilities come to mind:
 - a. An anchorage independent setting (3D growth, or perhaps organoids) might be a more relevant setting with which to compare the in vivo results. Would inhibition of cellular proliferation in such a setting compare better with the observed in vivo activity in MCC-13 vs the 2D setting?
 - b. The authors implicate H3Ser10 phosphorylation as a measure of the effectiveness of AURK inhibition. This phosphorylation is measurable as a PD marker in vivo. Does 2811 affect H3Ser10 phosphorylation status in the MCC-13 line in vivo? Why does inhibition of this phosphorylation in MCC-13 cells with treatment with AZD2811 not translate into cell size changes or into inhibition of proliferation?
6. The authors speculate that the inhibitors that they hoped to identify in the study may be used in combination with other agents (e.g. PD-L1, Bcl, and Topo II, etc). One presumes that such follow-up studies are either planned or in progress.

Grammar/typo issues noted:

1. Abstract: Comma is not needed between "AZD2811NP" and "reduced".
2. Bottom of second page of Background section: "nanoparticle based" should be hyphenated.
3. Second paragraph below section entitled "The AURKB inhibitor...": Comma is not needed between "AZD2811" and "was".
4. In the text and figure legends, sometimes there is a space between a value and the unit (30 nM), and in some cases, there is not (30nM). This should be standardized.
5. In Fig 4A, the graph shows an "n = ?".

Version 1:

Reviewer comments:

Reviewer #1

(Remarks to the Author)

New treatments would be welcome in Merkel cell carcinoma and data from the present report points to Aurora kinase B as a new and effective avenue of clinical intervention. The manuscript was previously reviewed extensively and the authors have done a fine job of answering all queries and questions to this reviewer's satisfaction. Where appropriate, new experimental data has been added and figure/figure legend clarified. The screening methodology, conducted in multiple cell lines, has been carried out with careful diligence and meets current standards of rigor and reproducibility in the field. The major claims are supported by the results and future experimental directions, such as drug synergy, are suggested. Overall, this is a well-written manuscript and will be of interest to all researchers and physicians in the field of Merkel Cell carcinoma. Recommend for publication.

Reviewer #2

(Remarks to the Author)

The authors have addressed all my concerns.

The results are still novel and important, although it has been 5 years since the previous submission.

I endorse publication of this revised manuscript.

Reviewer #3

(Remarks to the Author)

This revised manuscript, by Gelb, Garman, Brownell, and colleagues improves upon the original version that I reviewed some time ago. Indeed, such a time has elapsed that this was essentially a fresh read. As a drug repurposing screen, this paper presents what may be a directly clinically translatable finding, which is arrived at via two complementary methods, small molecule screening and druggable genome RNAi screening. While the identified target, Aurora Kinase B - having been worked for quite some time - is not especially novel in and of itself, the potential availability of clinically relevant molecules points to the potential for a high level of impact. The finding of MCC sensitivity to AurkB inhibition via two distinct methods of screening as well as the validation performed in vivo and in vitro raise confidence in the conclusions and support the use of AurkB inhibitors in MCC.

With this essentially fresh read, my notes indicate in a few places that exploration of viability in a 3D setting might also help to harmonize the differences with the in vivo response – and I see that I listed this in my initial review as well. I continue to think that a 3D-format viability assay, or perhaps use of VN-MCC organoids may help in the explanation as to why exceedingly modest potency against VN-MCC cells in a plate-based assay (Fig 2C) leads to in vivo activity in the same VN-MCC line (Fig 6 C, E).

Related to the above, I agree with the authors that more data were included in the revision to help detail differences in VP- vs VN-MCC line responses to AurkB inhibition. The included results suggest a more effective apoptotic response in VP-MCC, with variable response in VN-MCC lines. This is a match with the differential sensitivity of the lines to the inhibitor in the viability experiments. The authors further verify in new experiments that the biomarker effects on H3Ser10 phosphorylation as caspase-3 are stronger in the VP line than in the VN line. I agree that this explains why the in vivo response is stronger in the VP-MCC xenograft than the VN-MCC xenograft. But it remains curious why the compound induces an obvious and powerful cytostatic response and survival advantage in the VN-MCC line MCC13 in vivo at all, given the exceedingly modest response shown in vitro in Fig 3C and the reduced induction of apoptosis. The authors demonstrate that the concentration that the compound achieves in vivo exceeds the doses used in vitro, implying that an enhanced dose response running to a higher compound concentration relative to that shown in 3C may reveal a stronger effect. Implied as well in the flow cytometry results in Figure 5 is that an extended duration of treatment may realize an enhanced response in VN-MC cells. Dosing to such a higher concentration (to match or exceed the in vivo concentration attained) or at a longer timepoint (5day? 7day? 10 day) would be relatively simple experiments that could be done to help explain the still surprising in vivo results. Ultimately, however, the in vivo results shown suggest enhanced patient advantage pointing to the use of the compound in both MCC variants.

(minor point) I was unable to determine why Topo I inhibitors are specifically called out in Figure 1E. The corresponding text on pages 6 and 7 did not mention this at all, at least not that I could find.

Note that I was somewhat unexpectedly not able to access the supporting information that is frequently referred to in the text. Thus, I was unable to review the associated figures and legends, simply relying on the statements of the authors about this work and its relevance to the main points.

Dear Reviewers,

Thank you for reviewing our manuscript. After review of our original 2018 manuscript submission, we were invited to submit a revised manuscript addressing reviewer comments. However, due to circumstances stemming from personnel changes within our research team and then the COVID-19 pandemic, we were unable to submit a revised paper at the time.

We revived the project last year and have now addressed the reviewer comments which has improved the manuscript and made the results more compelling. Please find below point-by-point responses to the referee comments. In addition to changes made to address the comments, we have also made small changes to improve accuracy and readability.

Response to Reviewer #1 (Expertise: RNAi screen, functional genomics, Remarks to the Author):

Merkel cell carcinoma (MCC) is a deadly cancer with a highly unmet medical need. Here, the authors ran a screen of ~4000 drugs to assess of virus positive and virus negative MCC cell lines at two different concentrations of the compounds. In parallel, a RNAi screen was run; both modalities (re: small molecule and siRNA) identified Aurora Kinase B as a critical therapeutic target. This reviewer applauds the approach. Overall, the paper is very thorough, well-executed, with impressive data analysis and provides sufficient explanation and follow-up studies of the screening hit(s). The major drawback is lack of novelty, as explained in more detail below. As a point of comparison, note that a NatComm paper (Paulson et al) was just published this week regarding class I HLA and resistance to immunotherapy in MCC, which by contrast to the present work, was a finding quite unexpected and novel (and worthy of publication in this journal).

Thank you for applauding our experimental approach and for your positive feedback. We agree with the reviewer that the Paulson et al case series was an interesting finding with implications regarding resistance to cell-based immunotherapy. At the same time, the present study is also novel in advancing the understanding of MCC and providing important, well validated data about small molecule inhibitors and intersectional screening with functional genomics across multiple cell lines. Our identification of multiple top hits that are primed for drug repurposing and target development is even more relevant considering the resistance that can arise to immunotherapy, as reported by Paulson et al.

Specific Comments:

1. Background

a. Paragraph # 3 (starting with “Aurora kinase B...”) seems to be out of place, giving background information about the hit before describing the screen. It may be better positioned in the end of the “Background” section.

Thank you for pointing this out. We agree with you and have moved the paragraph to the end of the Background section.

b. Given the caliber of the journal, paragraph # 3 attests to the fact that the gene hit is not really novel (re: ref 16 and the tie-in to survivin pathway)

The fact that our top hit was associated with a proposed MCC target in prior work further validates our approach. We could have proceeded with the preclinical development of any of our top hits. We selected AURKB inhibitors because there is a nanoparticle formulation of AZD2811 in clinical development and therefore closer to clinical translation. As our goal is to

help patients, we choose to not exclude drugs because they target a complex that has previously been implicated in MCC. The novelty of our work includes the large scope of our screen, the inclusion of control cell lines, the orthogonal screening approaches, and degree of functional validation. We are reporting a highly validated data set that will be a useful resource for those studying MCC and neuroendocrine carcinomas more generally. We also identified a much more potent way to target the chromosome passenger complex in MCC than survivin inhibitors.

c. End of paragraph # 4 –worth stating that the cell lines are human. *This has been updated, thank you for pointing it out.*

2. Results

a. General note: All figures that have relevant information in supplementary material should have a reference to relevant supplementary figures in the legends. *We have updated all figure legends to include relevant supplementary figures throughout the manuscript. Thank you for this suggestion.*

b. Sub-Section: Identification of VP-MCC and VN-MCC-selective drug targets and compounds

i. Paragraph # 1 indicates that there were 99 VN-MCC selective hits, yet paragraph # 2 states that there were 156 initial target subcategories. How is this possible?

Thank you for pointing out this confusing aspect of our manuscript. Paragraph # 1 indicates that, compared to control cell lines, our screen identified 99 VP-MCC selective compounds and 15 VN-MCC selective compounds (Supplementary Fig. S2). As for grouping compounds by target subcategories, all 1912 mechanistically annotated library compounds (Supplementary Table S4) were grouped by target subcategory (n=156) and more broadly by compound class (n=56).

We grouped the VP-MCC and VN-MCC selective compounds in our screen by target subcategory, and this analysis helped us identify four target subcategories enriched among the VP-MCC selective agents, but no targets were selective for VN-MCC cells (Supplementary Fig. S4). We have clarified this analysis and the mechanistic grouping of compounds in our revised manuscript.

NB: The number of VP and VN selective compounds was the sum of the compounds identified in NPC and MIPE libraries. As topotecan was in both libraries, we have adjusted the number of VP-MCC selective compounds to 98 to account for this redundancy.

Were a lot of compounds non-specific?

We analyzed compounds based on their activity in the cell lines we tested. Compounds were classified as pan-active if they reduced viability in all 10 cell lines (control + MCC) (n = 227), control-selective compounds exclusively reduced viability of control cell lines (n = 28), and MCC-selective compounds exclusively reduced viability in MCC cell lines (n = 39) (Fig. 1B). The remaining compounds (n = 2252) had activity that was not specific to one of the three primary cell groups. We have revised the manuscript to explicitly state these findings.

Also, here it would be better to refer to compounds as drugs, not as “targets” as no compound will have absolute specificity for a given target.

Thank you for this suggestion, we have modified the text to avoid using the term target unless we are inferring mechanism from the annotated targets of the compounds.

c. Sub-Section: Follow-up small molecule screens support AURKs as a high priority

target for VP-MMC

i. Line # 3: were the listed AURK inhibitors included in the primary screen or are these additional compounds used for validation?

Thank you for pointing out this point of ambiguity. We have revised the Results to make it clear that we were re-testing the same compounds. All the compounds were validated in triplicate across all cell lines.

d. Sub-section AURK signaling partners in the CPC also maintain VP-MCC viability. Why are we not shown the individual knockdown data for the three proteins seen in the Chromosomal Passenger Complex (CPC) of which AURKB is a part of, that is, survivin, INCENP, and Borealin? It would help to substantiate the author's claims.

Our original study used an RNAi kinase library. We have now conducted a larger functional genomics screen using the Silencer Select Human Druggable siRNA Library with 10,418 genes. The new data includes knockdown of both survivin and INCENP and both are now included in supplementary Fig S8 demonstrating that knockdown significantly reduced VP-MCC viability but not VN-MCC viability. We continue to include data on the pharmacological inhibition of survivin (Supplementary Fig. S8A). Unfortunately, borealin was not included in any of our RNAi libraries.

Note: the tie-in to stathmin-1 and gastric/nasopharyngeal cancers is very perceptive.

Thank you.

e. Sub-section: RNAi screening independently identifies aurora kinase B as a MCC therapeutic target

i. Why were MKL-2 and MCC-26 lines chosen for the RNAi screen?

We have updated the results to clarify that MKL-2 and MCC26 cell lines were chosen for the RNAi screen because they had the highest transfection efficiency in our high-throughput system.

ii. If both VP-MCC and VN-MCC provide medical challenges (though in different percentages of patient population, why do the authors continuously select hits specific to VP lines that are necessarily excluded from VN?

Ideally, we would have found a class of compounds that was efficacious in VP-MCC and VN-MCC cell lines without showing toxicity in our control cell lines. However, as shown in Fig. 1E there were no compound classes that were significantly MCC selective over controls. As we note in the manuscript: "We focused on validating and developing drugs that selectively reduced VP-MCC viability because VP-MCC is more common than VN-MCC, the VP-MCC cell lines we used are more representative of native MCC tumors than the VN-MCC cell lines, and more VP-MCC candidate compounds were identified in the screen. In this approach we used the VN-MCC cell lines as closely-related controls/comparators to better identify targets specific to VP-MCC." The reviewer is correct that VN-MCC is a medical challenge and identifying new drugs to treat VN-MCC would be useful, this goes beyond the focus of the current study. In the future we plan to test additional compounds that share the same targets as the 15 compounds we identified as VN-MCC selective in our primary screen. Importantly, we ultimately discovered that AURKB inhibitors have activity against both subtypes of MCC, with higher potency and efficacy in VP-MCC. We have added text to the Discussion section concerning the limited conclusions we have made on VN-MCC in the current study.

f. Section: Inhibition of aurora kinase B reduces tumor burden and extends survival in preclinical mouse models of MCC

i. Paragraphs # 2 and 3: The authors show that animals injected with MKL-1 cells respond better to AZD2811NP than animals that were injected with MCC-13. This is

demonstrated by their longer life span (>200 days) compared to MCC-13 treated with AZD2811NP (~120 days). However, if you compare the placebo-treated groups, mice injected with MCC-13 die sooner than MKL-1-injected mice. Are MCC-13 tumors more aggressive? If so, the fold increase in survival (AZD2811NP/placebo mice) is roughly 2X in tumors from both cell lines.

Thank you for this astute observation. Yes, MCC13 xenograft tumors grow faster than MKL-1 xenograft tumors. However, consistent with having higher in vitro potency in VP-MCC cells, AZD2811NP increased median survival by 3.6-fold in MKL-1 mice (229 days relative to 63.5 days in the placebo group) versus 2.6-fold in MCC13 mice (84 days relative to 31.5 days in the placebo group). We have updated the results to include the proportionally larger fold-change improvement in the VP-MCC xenograft tumors.

3. Discussion

a. Paragraph # 4: Fig. 2A shows five drug target categories that ranked higher than AURK. Why did the authors decide to focus on this pathway?

We are engaged in multiple studies following up on target subcategories identified in our screen. We chose to focus this manuscript on the AURK pathway because there is a nanoparticle formulated AURK inhibitor in clinical trials providing a translational path to the clinic. Moreover, multiple AURK inhibitors reduced VP-MCC viability (suggesting an on-target effect) and targeting other members of the chromosomal passenger complex (e.g., survivin and INCENP) also reduced VP-MCC viability.

b. Paragraph # 4: Given the known efficacy of sepantonium bromide, have the authors tried to combine this treatment with AZD2811NP? Would they expect an additive/synergistic effect?

Thank you for this suggestion. We have not tested these two treatments in combination. We have added a sentence to this paragraph speculating that combining these two CPC inhibitors may be effective in VP-MCC.

c. 2nd to last paragraph: If AZD2811NP is currently being tested for small cell lung cancer, would these findings be considered sufficiently novel?

The fact that AZD2811NP has shown clinical promise in another neuroendocrine cancer is one of the reasons we decided to pursue its preclinical development. Our novel findings being validated by results in another related cancer strengthens the study. There is no a priori way to predict which drugs will work in related cancers, and unbiased large-scale screening approaches such as ours is a primary means of discovering relevant therapeutic targets.

4. Figure 2

a. Why were VN-MCC targets actively avoided?

Drugs that reduced the viability of VN-MCC were not avoided. Although we identified 15 compounds that were VN-MCC selective, there were no drugs that shared target subcategories or compound classes. This was true when comparing against control cell lines (Fig. 1) and when comparing to VP-MCC (Fig 2). As we state in the results "...but no targets were selective for VN-MCC cells (Supplementary Fig. S4)."

b. The divergence of selected categories from the diagonal line is quite weak.

We appreciate that this way of visualizing aggregated dose responses is not commonplace and interpreting distances is not intuitive. This is why we highlighted the target subcategories and classes that demonstrated a meaningful and statistically significant difference. Our cutoffs are actually quite stringent. This fact is reflected by the very dramatic differences observed when comparing the dose response curves of the individual drugs that make up the target

subcategory or class (e.g., Fig 3C or Supplementary Fig S6)

5. Figure 3

a. Fig. 3c -Compared to other compounds shown in Fig. S6, Tozasertib shows a relatively weak response in VP-MCC cell lines. Why was it chosen for the main figure? If it is included in order to show the range of responses, it should be stated in the text.

Thank you for pointing this out. Initially, we chose tozasertib for the main figure because there was no publicly available data on AZD2811, whereas tozasertib data was available in The Genomics of Drug Sensitivity in Cancer (GDSC) Database. In the revised manuscript, we replaced tozasertib with AZD2811 in Fig. 3D. The AZD2811 IC₅₀ values were obtained from publicly available data from the Dependency Map project, which allowed us to compare AZD2811 effects across 246 cancer cell lines. We have now moved the tozasertib data to Figure S6C alongside GSK-1070916A. Excitingly, AZD2811, tozasertib, and GSK-1070916A all demonstrated greater potency in VP-MCC cell lines compared to hundreds of other human cancer cell lines.

6. Figure 4

a. One of the reasons the authors chose AURKB, rather than other isoforms, was due to the difference in efficacy of its knock-down on VP vs. VN cell line viability. However, based on Fig. 4E, the response in MCC-26 cell line is not representative of other VN cell lines. Why did the authors choose MCC_26 cell line to determine their hits?

As discussed in the response to Reviewer 1, Section 2, Comment e, we used MKL-2 and MCC26 for our RNAi screens as they have the highest transfection efficiency. Moreover, as discussed in the response to Reviewer 1, Section 2, Comment d, we have replaced our original RNAi kinome screen with a larger druggable genome screen. The new results continue to show that MCC26 has less of a reduction in viability than MCC13 and UISO VN-MCC cell lines. The differences among the VN-MCC cell lines are consistent with the inter-cell line heterogeneity observed in how these cell lines respond to ADZ2811 treatment. Our follow up studies now demonstrate that AZD2811 is active against MCC26, but it requires a longer duration of treatment compared to other cell lines (72 hours versus 24 hours). In MCC26, AZD2811 reduced H3Ser10 phosphorylation and increased polyploidy (Fig. 5), and increased cell size, multinucleation, micronucleation, and centrosome amplification (Supplementary Fig. S10).

7. Figure 3D and Figure S6c

a. Fig. 3D contains information from 420 cell lines from GDSC, whereas Fig. S6c contains 950 cell lines. Why is there such a large difference?

We do not know how the GDSC curates their public data sets and we cannot answer as to why the GDSC contains different numbers of data points for different compounds.

Response to Reviewer #2 (Expertise: MCC, therapy, Remarks to the Author):

In this manuscript by Gelb et al, they screened nearly 4000 drugs to search for potential chemotherapeutic drugs for Merkel cell carcinoma (MCC). Their AUC analyses combined with principal component analysis, hierarchical clustering, and bubble plot analysis identified that small molecules targeting AURK, BCL-xL, FLT, HDAC, and topo II selectively suppress the viability of Merkel cell polyomavirus (MCPyV)-positive(VP)-MCC. RNAi kinome screening further confirmed that AURKB inhibition is the most effective in selectively reducing the viability of VP-MCC cells. They also showed the importance of AURKB downstream signaling for VP-MCC cell survival by knocking down AURKB signaling partners, including Survivin and INCENP as well as AURKB substrate Stathmin-

1, which also reduces MCPyV(+) MCC cell viability. Consistent with the inhibition of mitotic kinase, reduction in MCC cell viability by AURKB specific inhibitor AZD2811 was associated with cell cycle arrest in G2M. Finally, by using a xenograft mouse model, the authors showed that nanoparticle-formulated AZD2811NP inhibits the xenograft growth of both MKL-1 and MCC-13. MCC is the 2nd most common cause of skin cancer death, next to melanoma. Prior to the discovery of MCPyV within MCC, cancer cell signaling pathways that could be targeted for MCC chemotherapy had been unknown. This manuscript demonstrated that the AURKB inhibitor selectively inhibits VP-MCC by unbiased screening method. If the AURK pathway is activated by MCPyV and is critical in virus-induced MCC genesis, AURKB is the rational chemotherapeutic target, and the AURKB inhibitor may be promising for future clinical trials.

General comments:

Overall, the manuscript is well-written, and the conclusion is supported by the presented results. One missing component, however, is the mechanistic insight into the selective inhibition of MCPyV(+) MCC cancers. Additionally, the cellular mechanism that decreased viability by the AURKB inhibitor is unclear.

Thank you for your careful review and for your encouraging comment regarding our work and the promise of AZD2811NP in future clinical trials.

We agree that increased mechanistic understanding of AZD2811 in MCC is important, and we have expanded our mechanistic experiments to elucidate the commonalities and differences between VP-MCC and VN-MCC responses (see Figs. 5, S10, S11).

Specific comments:

1. In the paper, authors provide one example that viral activation of Stathmin-1 activation may explain AURKB-dependence of MCPyV(+) MCC by showing higher Stathmin-1 expression in VP-MCC. Does Merkel cell virus oncogene such as viral large T or small T expression affect AURKB expression?

Thank you for this question. We have not addressed this directly. However, because there is no significant difference in AURKB expression levels between VP- and VN-MCC in the negative control transfected conditions (Fig. 4C), we hypothesize that Merkel cell polyomavirus oncogenes do not affect AURKB expression.

2. Is AURKB highly activated in VP-MCC?

We have not measured this directly, but this manuscript has shown that AURKB activity is necessary for VP-MCC viability, demonstrating its potential as a therapeutic target. Based on our new data in the revised manuscript, we believe VN-MCC cell lines are less sensitive to AURKB inhibition because they exist in a state of mitotic dysregulation and polyploidy/aneuploidy at baseline.

3. Authors focus on decreased viability induced by AURKB inhibition in WST assays. Is decreased viability due to mitotic arrest or cell death induced by apoptosis or mitotic catastrophe? While results in Fig. 5C suggest G2/M cell cycle arrest is due to decreased viability, authors need to clarify whether drug treatment is associated with cell death both in vivo and in vitro.

Based on these comments and questions, we examined the mechanism of AZD2811- induced cell death. We were able to show AZD2811 induced robust apoptosis in VP-MCC (WAGA and MKL-1) and a weak (MCC26) or no apoptosis (MCC13) in VN-MCC (Supplementary Fig. S11). We also show AZD2811 induces mitotic dysregulation in all MCC cell lines including G2/M

arrest (WAGA, MKL-2, MKL-1, MCC13) (Fig. 5C, D), increased polyploidy (MCC26) (Fig 5C, D), and mitotic catastrophe (WAGA, MKL-1, MCC26) (Supplementary Fig. S10).

In addition, we have analyzed tumor slides from in vivo AZD2811NP xenograft studies. We did new animal studies using established xenograft tumors treated for only 10 days before tissue collection. AZD2811NP-treated VP-MCC (MKL-1) tumors had small volumes with no viable tissue. VN-MCC (MCC13) xenograft tumors, as shown in the new Supplementary Fig. S14, treated with AZD2811NP showed reduced phospho-Histone H3 and Ki67 expression demonstrating cell cycle arrest and reduced cell proliferation. There was no difference in cleaved caspase 3 (an indicator of apoptosis) between AZD2811NP- and vehicle-treated xenografts. All these results are consistent with our in vitro studies.

4. VP-MCC cell lines used in each assay are not consistent. In the AURKB knockdown analysis, WaGa and MKL-1 are used for viability evaluation (Fig. 4E), while WaGa and MKL-2 are used in AZD2811 drug assays and MKL-1 is used in xenograft assays. At the very least, MKL-1 needs to be included in all the assays.

We agree that consistency across cell lines using different experimental readouts is critical for interpretation. WAGA, MKL-1, and MKL-2 cell lines were all used in the high-throughput drug screen and validation. Additionally, we tested the effect of AURKB knockdown on viability in all three of these VP-MCC cell-lines (MKL-2 shown in Fig. 4A, B and WAGA, MKL-1 shown in Fig. 4C-E). In our revised experiments, we were able to include MKL-1 in our phospho-Histone H3 Western blot (Fig. 5A, B). We also included MKL-1 in our mechanistic studies such as caspase Glo and cleaved PARP Western blots (Supplementary Fig. S11) and mitotic catastrophe (Fig. S10). For ethical and cost reasons, only one VP and one VN MCC cell line could be used in the xenograft mouse studies.

5. Viral T antigen expression in VP-MCC has been shown to be required for MCC cell growth. Does AZD2811 affect viral T antigen expression?

Thank you for this question. We have treated three VP-MCC cell lines (WAGA, MKL-1, and MKL-2) with 3nM and 30nM AZD2811 and there was no difference in LT expression at 24 hours as shown in the new Supplementary Fig. S12.

Response to Reviewer #3 (Expertise: Drug screening, Remarks to the Author):

This manuscript, by Gelb, Brownell, and colleagues, describes screens of known agents and RNAi targets that converge on the identification of the potential utility of Aurora B kinase inhibitors as therapeutic options in Merkel cell carcinoma. Since the small agent most fully profiled (AZD2811-NP) is in use clinically, this work offers a clear path toward clinical translation – a significant impact. Overall the work is quite complete and follows through from in vitro screening to in vivo results. Further, the paper is generally well written, with a bare minimum of typographical and grammatical errors. This work is likely suitable for publication in Nature Communications, but there are a few scientific questions and comments in this reviewer’s mind that need to be addressed:

Thank you for your careful review of this paper and for pointing out the clear path toward clinical translational impact based on the results.

1. In the abstract, as well as occasionally in the text, the authors state that this work identifies Aurora B as a novel target. This is a bit of an overstatement, since the kinase is well established as a cancer drug discovery target, and clinical stage agents exist. The statement should always be qualified with “in Merkel cell carcinoma”.

Thank you for pointing this out and for acknowledging the novelty in MCC. We have updated the manuscript accordingly.

2. The 156 target subcategories could use some additional brief explanation, especially given the lack of associations with the inhibition phenotypes. How granular are the target designations (e.g., all kinases inhibitors in one class vs sufficient target detail to separate out different kinases or branches of the kinome, etc.)? In addition to this, most agents, likely including many of those used in this study, have multiple molecular targets. How annotated are the target classes for each compound? It is entirely conceivable that one compound may fit into more than one subcategory; a slightly different grouping with more such detail might significantly change the lack of associations (or lack of associations) observed.

Thank you for this insightful comment. We agree that for our data to be of optimal use requires an understanding of the compound classifications. We have now included Supplementary table S4 annotating the target subcategories. Target annotations of the library compounds were previously published [Oncotarget. 2015;6(34):35247.]

3. In the results section describing the follow-up screening with small molecules (Fig. 3 and S6), it is stated that modest effects were observed at high concentrations with certain Aurora inhibitors. However, both figure 3C and S6A, some of the small molecules appear to show full inhibition of at least some of the VN-MCC lines. While the potencies are less than in the VP-MCC lines on average, the full efficacy observed at higher concentrations for many of them does not seem modest.

Thank you for pointing this out. We agree and have updated the text accordingly.

4. The most confusing aspect of the work is the disconnect between some of the results. For example, in the small molecule target subcategories work (Fig. 2 and attendant discussion), Aurora inhibitors are shown to be selective for VP-MCC vs VN-MCC. This notion is backed up by cellular work with AURKB inhibitor molecules (Fig. 3 and discussion), and also by a portion of the RNAi results, in which AURKB knockdown is ineffective in at least one VN-MCC line. However, the extended sampling of RNAi against AURK in additional MCC lines points toward the MCC-26 line being a bit of an outlier, rather than representative of VN-MCC as a whole. This seems to indicate that AURKB inhibition is not as selective for VP- vs VN-MCC as claimed earlier in the text.

We agree with the reviewer's comment. In the revised manuscript we now include additional experimentation that elucidates the similarities and differences in how VP and VN-MCC respond to AURKB inhibition, see response to Reviewer 2, Comment 3. The new results also explain observed inter-cell line differences among the VN-MCC cell lines, see response to Reviewer 1, Comment 6. In brief, all MCC cell lines showed mitotic dysregulation and cell cycle arrest after treatment with the AURKB inhibitor AZD2811 whereas there were differences in the rate, magnitude, and mechanism of subsequent cell death. We think the new data answer many of the confusing results of the study.

5. Related to point 4 above, AZD2811 is indeed less effective in inhibition of proliferation in the VN-MCC cell lines, including MCC-13. However, AURKB knockdown with RNAi is effective in MCC-13, and AZD2811 is shown to work in vivo in an MCC13 derived xenograft. The authors offer that changes in cellular size are related to the differing effectiveness of the molecule, but this explanation seems not wholly sufficient in light of the in vivo activity of AZD2811 in MCC13 (which was resistant to changes in cellular size). Are there other underlying reasons for the differences? A few additional possibilities come to mind:

As alluded to in the above response (Reviewer 3, Comment 4), our revised manuscript now reports on mechanistic differences among the MCC cell lines. We now show that at baseline MCC13 cells already demonstrated enlarged cell size, micronucleation, and multinucleation with centrosome amplification, suggesting these cells exist in a state of mitotic dysregulation. AZD2811 treatment did not increase the percentage of these enlarged cells but did cause subjective nuclear deformities and non-apoptotic cell death (Fig. 3, Supplementary Figs. S10, S11)

a. An anchorage independent setting (3D growth, or perhaps organoids) might be a more relevant setting with which to compare the in vivo results. Would inhibition of cellular proliferation in such a setting compare better with the observed in vivo activity in MCC13 vs the 2D setting?

We appreciate the suggested experiment to test the hypothesis that anchorage independent growth might be contributing to differences observed in MCC13 cells. However, considering the new mechanistic data, our findings are now more harmonized, and we feel additional experimentation to test alternative hypotheses are no longer required.

b. The authors implicate H3Ser10 phosphorylation as a measure of the effectiveness of AURK inhibition. This phosphorylation is measurable as a PD marker in vivo. Does 2811 affect H3Ser10 phosphorylation status in the MCC13 line in vivo?

Thank you for raising this point. We now report on the reduction in H3Ser10 phosphorylation in MCC13 xenografts in the Results section (Supplementary Fig. S14A).

Why does inhibition of this phosphorylation in MCC13 cells with treatment with AZD2811 not translate into cell size changes or into inhibition of proliferation?

As we outlined in the above responses to this Reviewer comment, our new data demonstrate that MCC13 cells show enlargement and mitotic dysregulation at baseline and AZD2811 treatment does not change average cell size (Supplementary Fig. S10) but does reduce H3Ser10 phosphorylation, induce cell cycle arrest (Fig. 5), and decrease viability (Fig. 3).

6. The authors speculate that the inhibitors that they hoped to identify in the study may be used in combination with other agents (e.g., PD-L1, Bcl, and Topo II, etc). One presumes that such follow-up studies are either planned or in progress.

The reviewer is correct. We have ongoing synergy combination studies that are beyond the scope of the current report.

Grammar/typo issues noted:

Thank you, these have been corrected.

1. Abstract: Comma is not needed between “AZD2811NP” and “reduced”.

2. Bottom of second page of Background section: “nanoparticle based” should be hyphenated.

3. Second paragraph below section entitled “The AURKB inhibitor...”: Comma is not needed between “AZD2811” and “was”.

4. In the text and figure legends, sometimes there is a space between a value and the unit (30 nM), and in some cases, there is not (30nM). This should be standardized.

5. In Fig. 4A, the graph shows an “n = ?”.

Thank you for reviewing our revised manuscript. We appreciate the opportunity to submit this second revision wherein we address the latest reviewer comments. Please see below for point-by-point responses to the reviewer comments.

Reviewer #1:

New treatments would be welcome in Merkel cell carcinoma and data from the present report points to Aurora kinase B as a new and effective avenue of clinical intervention. The manuscript was previously reviewed extensively and the authors have done a fine job of answering all queries and questions to this reviewer's satisfaction. Where appropriate, new experimental data has been added and figure/figure legend clarified. The screening methodology, conducted in multiple cell lines, has been carried out with careful diligence and meets current standards of rigor and reproducibility in the field. The major claims are supported by the results and future experimental directions, such as drug synergy, are suggested. Overall, this is a well-written manuscript and will be of interest to all researchers and physicians in the field of Merkel Cell carcinoma. Recommend for publication.

Thank you for your thoughtful and positive feedback. We are pleased that our revisions have addressed the previous concerns and improved the manuscript.

Reviewer #2:

**The authors have addressed all my concerns.
The results are still novel and important, although it has been 5 years since the previous submission.
I endorse publication of this revised manuscript.**

Thank you for the positive and encouraging comments, your suggestions helped us improve the study.

Reviewer #3:

This revised manuscript, by Gelb, Garman, Brownell, and colleagues improves upon the original version that I reviewed some time ago. Indeed, such a time has elapsed that this was essentially a fresh read. As a drug repurposing screen, this paper presents what may be a directly clinically translatable finding, which is arrived at via two complementary methods, small molecule screening and druggable genome RNAi screening. While the identified target, Aurora Kinase B - having been worked for quite some time - is not especially novel in and of itself, the potential availability of clinically relevant molecules points to the potential for a high level of impact. The finding of MCC sensitivity to AurkB inhibition via two distinct methods of screening as well as the validation performed in vivo and in vitro raise confidence in the conclusions and support the use of AurkB inhibitors in MCC.

With this essentially fresh read, my notes indicate in a few places that exploration of viability in a 3D setting might also help to harmonize the differences with the in vivo response – and I see that I listed this in my initial review as well. I continue to think that a 3D-format viability assay, or perhaps use of VN-MCC organoids may help in the explanation as to why exceedingly modest potency against VN-MCC cells in a plate-based assay (Fig 2C) leads to in vivo activity in the same VN-MCC line (Fig 6 C, E).

Related to the above, I agree with the authors that more data were included in the revision to help detail differences in VP- vs VN-MCC line responses to AurKB inhibition. The included results suggest a more effective apoptotic response in VP-MCC, with variable response in VN-MCC lines. This is a match with the differential sensitivity of the lines to the inhibitor in the viability experiments. The authors further verify in new experiments that the biomarker effects on H3Ser10 phosphorylation as caspase-3 are stronger in the VP line than in the VN line. I agree that this explains why the in vivo response is stronger in the VP-MCC xenograft than the VN-MCC xenograft. But it remains curious why the compound induces an obvious and powerful cytostatic response and survival advantage in the VN-MCC line MCC13 in vivo at all, given the exceedingly modest response shown in vitro in Fig 3C and the reduced induction of apoptosis. The authors demonstrate that the concentration that the compound achieves in vivo exceeds the doses used in vitro, implying that an enhanced dose response running to a higher compound concentration relative to that shown in 3C may reveal a stronger effect. Implied as well in the flow cytometry results in Figure 5 is that an extended duration of treatment may realize an enhanced response in VN-MC cells. Dosing to such a higher concentration (to match or exceed the in vivo concentration attained) or at a longer timepoint (5day? 7day? 10 day) would be relatively simple experiments that could be done to help explain the still surprising in vivo results. Ultimately, however, the in vivo results shown suggest enhanced patient advantage pointing to the use of the compound in both MCC variants.

(minor point) I was unable to determine why Topo I inhibitors are specifically called out in Figure 1E. The corresponding text on pages 6 and 7 did not mention this at all, at least not that I could find.

Note that I was somewhat unexpectedly not able to access the supporting information that is frequently referred to in the text. Thus, I was unable to review the associated figures and legends, simply relying on the statements of the authors about this work and its relevance to the main points.

We greatly appreciate the reviewer's thoughtful comments and suggestions. Addressing these have allowed us to further improve the manuscript. As the reviewer's comments addressed three topics, we have responded to each topic below.

- 1. discrepancy between the in vitro and in vivo response to AZD2811 in the VN-MCC cell line, MCC13**

We appreciate the reviewer's concern regarding the discrepancy between the in vitro and in vivo response to AZD2811 in MCC13 cells. As the reviewer astutely suggests, higher drug concentrations and longer duration of treatment could both account for the apparent discrepancy. Following the reviewer's suggestion, we tested both hypotheses and found them to be true. We treated MCC13 and MCC26 VN-MCC cells with DMSO or AZD2811 at concentrations of 30 nM, 300 nM, and 1 μ M for 6 days. For both cell lines, AZD2811 showed higher potency and efficacy when assessed at 6 days. The enhanced effect of AZD2811 on VN-MCC viability with extended treatment data are now included in the revised results and are shown in **Supplementary Fig. S6A/B**. These findings show that with prolonged treatment times, AZD2811 has congruent in vitro and in vivo responses in MCC13 cells.

MCC13 cells were noted to have shown minimal increases in cell size and centrosome amplification after 3 days of treatment with AZD2811. Having shown more viability reduction with longer treatment in MCC13 cells, we also quantified relative cell size and the percentage of multicentrosomal cells after 6 days. Both were significantly elevated. These findings are presented in **Supplementary Fig. S10D/E**. Increasing the treatment time and dose did not, however, induce apoptosis in MCC13 cells (**Supplementary Fig. S11A**). These new findings have been added to the results section.

We thank the reviewer for suggesting the use of 3D culture to explore the discrepancy between in vitro and in vivo responses to AZD2811 in MCC13 cells. We did attempt a soft agar colony formation assay using both MCC13 and MCC26 cell lines. However, technical challenges prevented us from producing consistent or interpretable results. Considering the positive results obtained from the longer duration treatment, we decided to cease attempting 3D culture experiments as the discrepancy had been resolved.

2. Topo I Inhibitors in Figure 1E

Thank you for pointing out the lack of reference to Topo I inhibitors in the manuscript. As the annotation is not referenced in this work, we have removed it from **Figure 1E**.

3. Supporting Information Accessibility

We regret that the reviewer had difficulty accessing the supporting information (Supplementary Datasets). We verified with the journal that the files were properly uploaded and included with the reviewer materials. We hope this issue was resolved. We have included the revised Supplementary files with this submission.